# Source contribution to ozone pollution during June 2021 fire events in Arizona: Insights from WRF-Chem tagged O$_3$ and CO

Yafang Guo[1,2], Mohammad Amin Mirrezaei[1], Armin Sorooshian[1,3], Avelino F. Arellano[1,3]

[1] *Department of Hydrology and Atmospheric Sciences*, *The University of Arizona, Tucson, AZ,*
*USA*

[2] *Sonoma Technology, 1450 N. McDowell Blvd., Suite 200, Petaluma, CA, USA*

[3] *Department of Chemical and Environmental Engineering*, *The University of Arizona, Tucson,*
*AZ, USA*

*Corresponding to*: Avelino F. Arellano (afarellano@arizona.edu)

**Abstract.** This study quantifies wildfire contributions to O$_3$ pollution in Arizona, relative to local and regional emissions. Using WRF-Chem with O$_3$ and CO tags, we analyzed emissions during June 2021, a period of drought, extreme heat, and wildfires. Our results show that background O$_3$ accounted for ~50% of total O$_3$, while local anthropogenic emissions contributed 24-40%, consistent with recent estimates for Phoenix. During peak smoke conditions, fire-related O$_3$ ranged
from 5 to 23 ppb (5-21 % of total O$_3$), averaging 15 ppb (15%). These estimates were compared with model sensitivity tests excluding fire emissions, which confirmed the spatiotemporal pattern of fire-driven O$_3$, though the model underestimated the magnitude by a factor of 1.4. The results further demonstrate that wildfires exacerbate O$_3$ exceedances over urban areas. Our analysis reveals key differences in O$_3$ sources: Phoenix's O$_3$ was mainly driven by local emissions, while
Yuma's was heavily influenced by transboundary transport from California and Mexico. Wildfires not only boosted O$_3$ formation but also altered winds and atmospheric chemistry in Phoenix and downwind areas. O$_3$ increases along the smoke plume resulted from NO$_X$-VOC interactions, with fire-driven O$_3$ forming in NO$_X$-limited zones near the urban interface. Downwind, O$_3$ chemistry shifted, shaped by higher NO$_X$ in central Phoenix and more VOCs in suburban and rural areas.
Winds weakened and turned westerly near fire-affected areas. This study highlights the value of high-resolution modeling with tagging to disentangle wildfire and regional O$_3$ sources, particularly in arid regions, where extreme heat intensifies O$_3$ pollution, making accurate source attribution essential.

## 1. Introduction

Ozone ($O_3$) pollution remains a pressing environmental and public health concern, especially in regions prone to wildfire activity (Jaffe et al., 2020; Jaffe et al. 2018; Jaffe and Wigder, 2012; Abatzoglou and Williams, 2016; David et al., 2021). Elevated $O_3$ levels can lead to a range of respiratory issues, cardiovascular problems, and other health complications, underscoring the importance of identifying and mitigating the sources of $O_3$ exceedances (e.g., Turner et al., 2016; Adhikari and Yin, 2020; Huangfu and Atkinson, 2020). Wildfires play a significant role in urban $O_3$ formation by emitting large quantities of volatile organic compounds (VOCs) and nitrogen oxides ($NO_X$), key precursors to $O_3$ production (Andreae, 2019; Akagi et al., 2012). These pollutants can travel long distances, combining with local emissions to exacerbate urban air quality issues (Xu et al., 2021; Jin et al., 2023; Jaffe and Wigder, 2012; Ninneman and Jaffe, 2021). In addition to $O_3$, wildfire smoke leads to an increase in other atmospheric oxidants, such as hydroxyl radicals (OH) and hydroperoxyl radicals ($HO_2$), while reducing $NO_2$ photolysis rates due to the shading effect of smoke plumes (Buysse et al., 2019). This shading effect reduces the amount of sunlight available for the photolysis of $NO_2$, which is a crucial step in $O_3$ formation. During wildfire events, stagnant air conditions often prevail in urban regions, preventing the dispersion of pollutants and allowing them to accumulate. For example, temperature inversions, which are more common during these events, trap pollutants near the ground, leading to higher concentrations of $O_3$ and other harmful substances (Alonso-Blanco et al., 2018; Burke et al., 2023; Jaffe et al., 2020; Pan et al., 2022; Xu et al., 2021). Besides $O_3$, smoke from fires contain large amounts of carbon monoxide (CO). During a wildfire, the high temperature during its flaming phase causes rapid oxidation of carbon-containing materials, but not all the carbon is fully oxidized to carbon dioxide ($CO_2$) during its (lower temperature) smoldering phase (Yokelson et al., 2003; Urbanski et al., 2008). CO emissions from wildfires can have far-reaching impacts, as CO is a gas with relatively medium lifetime in the atmosphere (ranges from several weeks to a few months). This is facilitated as well by associated plume rise especially during the fire's flaming phase.

Case studies, such as during various episodes of California wildfires, have demonstrated significant increases in urban $O_3$ levels, affecting cities far from the fire areas (Xu et al., 2021; Jin et al., 2023; Mcclure and Jaffe, 2018). During wildfire seasons, the complex interplay between local (urban) emissions, wildfire smoke, and meteorological factors contributes to significant $O_3$

exceedances, posing risks to both human health and ecological systems (Jaffe and Wigder, 2012; Jaffe et al., 2013; Selimovic et al., 2020; Holder and Sullivan, 2024). Identifying and disentangling the specific contributions of various emission sources to $O_3$ pollution during wildfire events is crucial for developing effective air quality management strategies. Regional and local $O_3$ levels are influenced not only by local production but also by regional and long-range transport of $O_3$ and its precursors. Common sources of $NO_X$ and VOCs—critical precursors to $O_3$—include fossil fuel combustion from vehicles, industry, and power plants, as well as natural biogenic emissions. However, when wildfires inject additional $NO_x$ and VOCs into the atmosphere, the overall levels of ground-level $O_3$ can rise exacerbating urban pollution as these plumes penetrate into city environments (e.g., Pfister et al., 2006; Brey and Fischer, 2015). This is especially the case in already $O_3$ non-attainment designated urban areas where fires increasingly occur at its urban interface.

Source attribution techniques offer an alternative perspective to quantify the primary contributors to enhanced $O_3$ levels during smoky periods by identifying the contributions of specific sources and regions, such as anthropogenic, fire, and biogenic emissions, regional and international transport and stratospheric transport. In general, there are two main modeling approaches for $O_3$ source attribution or source apportionment: 1) model sensitivity experiments; 2) species tagging methods ( Wang et al., 2009; Kwok et al. 2015; Clappier et al., 2017; Mertens et al., 2018; Butler et al., 2018; Thunis et al., 2019; Mertens et al., 2020). The latter modeling approach tracks $O_3$ formation by tagging precursors from particular source types and areas throughout the model simulation, providing a direct attribution of modeled $O_3$ levels to these sources. The tagging technique entails modifying the model's source code to incorporate tracers into the chemistry mechanism. Models of atmospheric chemistry and transport that have implemented a tagging technique to perform $O_3$ source attribution include among others the Community Multiscale Air Quality (CMAQ) model with a new version of Integrated Source Apportionment Method (ISAM) (De La Paz et al., 2024; Shu et al., 2023), a submodel called TAGGING in the EMAC (European Centre for Medium-Range Weather Forecasts – Hamburg (ECHAM)/Modular Earth Submodel System (MESSy) (Grewe et al., 2017) and MESSy-fied ECHAM and COSMO models nested n times or MECO (n) system (Kilian et al., 2024), the global Model for Ozone and Related chemical Tracers (MOZART- 4) (Emmons et al., 2012; Guo et al., 2017), the Nested Air Quality Prediction Model System (NAQPMS) (Zhang et al., 2020), CAM4-chem (Community Atmosphere Model

version 4 with chemistry) within the Community Earth System Model (CESM)  (Butler et al., 2018; Butler et al., 2020; Li et al., 2023; Nalam et al., 2024), the University of California Davis/Caltech air quality model (Zhao et al. 2022), the global chemical transport model (CTM) with assimilated meteorological observations from the Goddard Earth Observing System (GEOS-Chem) (e.g., Wang et al., 1998; Zhang et al., 2008; Whaley et al., 2015), and the Weather Research
and Forecasting model coupled with Chemistry (WRF-Chem) (e.g., Pfister et al., 2013; Gao et al. 2016; Lupaşcu and Butler, 2019; Lupaşcu et al., 2022; Romero-Alvarez et al., 2022).

Typical model sensitivity analysis to determine the impact of a particular process (like fire) on target variables (like $O_3$) are usually conducted in practice as a suite of process-denial experiments and/or a series of model simulations with brute-force incremental changes on particular parameters
or input datasets, along with developing model forward (decoupled direct method) and backward (adjoint) algorithms for sensitivity to emission calculations. Here, differences in simulated $O_3$ with and without fire emissions is interpreted to be the contribution of fire to modeled $O_3$ abundance. Unlike the tagging method, it utilizes the current model as is, without needing modifications. Models (and algorithms) that are used to predict how $O_3$ responds to changes in specific sources
of emissions include among others those using WRF-SMOKE-CAMx (SMOKE: Sparse Matrix Operator Kernel Emissions model; CAMx: Comprehensive Air quality Model with extensions) (Zhang et al., 2017, Goldberg et al. 2016), WRF-Chem (Li et al., 2015), High-Order Decoupled Direct Method in Three Dimensions (HDDM-3D) (e.g., Cohan et al., 2005), CMAQ (Yeganeh et al., 2024; Collet et al., 2014, Hakami et al., 2007), STEM (Hakami et al., 2006), and climate
chemistry model (E39C) (Grewe et al., 2012; Grewe et al., 2010).

These recent model and algorithm developments have shown the importance of integrating sophisticated modeling approaches and comprehensive data analysis to help better inform policies aimed at reducing $O_3$ pollution and its associated health impacts. Building on the work of Lupaşcu and Butler (2019) and Emmons et al. (2012), this study employs the tagging technique within the
WRF-Chem modeling system to investigate the sources contributing to $O_3$ exceedances during a recent Arizona wildfire season, particularly examining the impacts of fires on $O_3$ levels. While past research has explored source attribution, few studies have examined wildfire-driven $O_3$ pollution in urban environments using a high-resolution (3-km), convective-permitting, coupled chemistry-meteorology model with tagging capabilities. This approach is particularly valuable in

regions like Arizona, where urban $O_3$ levels are driven not only by local emissions but also by a complex interplay of meteorology, climate, topography, wildfire activity, and interstate pollution transport. The south/southeastern region of Arizona, including Phoenix, sits within the Sonoran Desert, a semi-arid/arid environment influenced by the North American Monsoon, the surrounding Mogollon Rim's complex terrain, and frequent wildfires, such as the Wallow Fire (2011) and

Telegraph Fire (2021), which occur at the urban-wildland interface. Managing $O_3$ pollution in this setting is particularly challenging, as Phoenix has been designated a moderate non-attainment area in recent years, and wildfire contributions further complicate regulatory efforts. As fire activity continues to escalate, addressing its role in $O_3$ pollution becomes increasingly urgent for effective air quality management.

This study leverages an advanced modeling framework to untangle the contributions of wildfire emissions, local anthropogenic activities, and regional transport to urban $O_3$ pollution in Arizona. To assess the effectiveness of our tagging approach, we conduct sensitivity experiments using the WRF-Chem model, including a zero-out fire emissions scenario to isolate wildfire impacts. Our analysis focuses on June 2021, a period marked by compounding extreme events in the Southwest

U.S. During this time, Arizona and much of the western U.S. experienced an unprecedented heatwave (Osman et al., 2022; Lo et al., 2023; White et al., 2023), likely driven by persistent high pressure and severe drought conditions (Thompson et al., 2022). This extreme heat coincided with record-breaking wildfires across the region (Jain et al., 2024), intensifying air quality concerns. Arizona, in particular, faced a convergence of extreme heat, prolonged drought, multiple active

wildfires, and dangerously high $O_3$ levels, making it an ideal case study for understanding wildfire driven $O_3$ pollution and its broader urban air quality implications.

However, Arizona's distinct environmental and atmospheric conditions, including its arid and semi-arid landscapes, extreme heat, and limited precipitation, create unique challenges for assessing wildfire behavior, air quality, and atmospheric chemistry. The shift between the dry

summer and monsoon seasons plays a crucial role in $O_3$ chemistry, as monsoon moisture alters wildfire smoke dynamics and atmospheric processes (Greenslade et al., 2024). Unlike California, where wildfire emissions interact with a more extensive urban footprint, Arizona's pollution dynamics—particularly over Phoenix—are influenced by its geographic isolation, creating a "sky island" effect that amplifies the urban heat dome. This interaction between heatwaves, wildfire

smoke, and urban pollution presents distinct challenges in understanding how these factors drive $O_3$ exceedances.

In this study, we examine two cases where Phoenix experienced significant wildfire smoke intrusions, using WRF-Chem at convective-permitting scale to capture the compounding effects of heatwaves and wildfires on $O_3$ pollution while also accounting for the meteorological feedbacks of fire emissions. The findings provide valuable insights not only for Arizona but also for other arid regions worldwide, including parts of the Middle East, Australia, and northern Africa, where similar environmental conditions influence wildfire behavior and air pollution dynamics.

Moreover, this study aims to elucidate the impact of wildfire events on $O_3$ chemistry and meteorology, addressing key gaps that persist despite extensive wildfire research (Xu et al., 2021; Jaffe and Wigder, 2012). Fire plume chemistry is highly complex and unpredictable, making it difficult to model $O_3$ formation accurately (Rickly et al., 2023; Robinson et al., 2021). Meteorological factors, such as temperature, solar radiation, and boundary layer mixing, further complicate predictions, as their interactions with wildfire emissions are not fully understood (Buysse et al., 2019; Li et al., 2024). Existing photochemical models often struggle to capture wildfire-driven $O_3$ formation, largely due to incomplete emissions data and high sensitivity to changing atmospheric conditions (e.g., Nopmongcol et al., 2017). Additionally, inconsistencies in background $O_3$ estimates across models make it difficult to separate wildfire contributions from other sources (e.g., Jaffe et al., 2018). Fire aerosols introduce further complexity, sometimes suppressing $O_3$ through radiative effects while at other times enhancing its formation via chemical interactions (e.g., Ninneman and Jaffe, 2021; Jiang et al., 2012). The relationship between smoke and urban pollutants is also nonlinear—while $O_3$ and $PM_{2.5}$ levels often rise with smoke, $NO_X$ trends remain inconsistent, and $O_3$ does not always increase in proportion to particle pollution (e.g., Baylon et al., 2015). Even more concerning, satellite-based smoke detection is often unreliable, with resolution and cloud cover limitations causing significant underestimation of smoke events (Buysse et al., 2019). To address some of these challenges, this study presents a case study over Phoenix, Arizona, utilizing an improved regional modeling framework with tagging capabilities as earlier mentioned. We integrate in our analysis available ground-based data on $O_3$, $PM_{2.5}$, CO, and $NO_X$ along with remotely sensed data on $NO_2$, HCHO, and CO, and local surface meteorological observations (e.g., wind, temperature, humidity). Our modeling results provide

insights on how wildfires—both near and distant—contribute to ozone production in Phoenix and how smoke alters local meteorological conditions, particularly wind patterns. By refining our understanding of these interactions, this study advances efforts to improve $O_3$ prediction, air quality management, and wildfire impact assessment in fire-prone urban regions.

This paper is organized as follows: Section 2 presents the observational datasets from ground-based Environmental Protection Agency's Air Quality System (EPA AQS) sites and satellites, alongside the model setup. Section 3 begins with a detailed introduction of the selected cases, followed by an analysis of comprehensive $O_3$ source apportionment. The discussion and summary are presented in Section 4.

## 2. Data and Methods

### 2.1 Study region and period

As mentioned, this is a case study focusing on 2021 dry summer (June) in Arizona, where most of the state falls under the arid/semi-arid region. This period was notably severe, exacerbated by a combination of prolonged drought and an intense heatwave. The state experienced one of its hottest Junes on record, with temperatures frequently exceeding 115°F (46°C), with no significant precipitation recorded. This extreme heat, combined with exceptionally dry conditions, were pivotal in the ignition and spread of multiple wildfires, leading to numerous large wildfires. In total, dozens of fires were reported across Arizona and New Mexico during this period, many sparked by lightning strikes on desert landscapes.

The Telegraph Fire, one of the largest wildfires in Arizona's history, began on 4 June 2021, near Superior, Arizona. By the time it was fully contained on 3 July 2021, the fire had burned over 180,000 acres. The burn area was located in the southernmost region of Tonto National Forest, primarily characterized by desert shrubs and grassland vegetation (USDA, 2024). The Rafael Fire started on 18 June 2021 to the southwest of Flagstaff, which prompted widespread evacuations and road closures. The Rafael Fire had burned over 38 square miles by late June in the Coconino National Forest, where evergreen shrubs were the dominate vegetation type (Conservation Biology Institute, 2024).

Figure 1 provides a comprehensive overview of the study region, focused on the Phoenix-Mesa-Scottsdale metropolitan area in Arizona, with Phoenix as the principal city. Panel (a) presents a

topographic map with red dashed lines outlining the metropolitan area. Phoenix is situated in Salt River valley within the Sonoran Desert, surrounded by small mountain reliefs. To the far northeast is the steep mountain ranges (Mogollon Rim) which forms the southern edge of the Colorado Plateau (Figure 1a). The built environment in the Phoenix Metro contributes to an urban heat island effect, which, along with thermo-topographical circulation, influences wind flow pattern across the region that is mostly westerly during the day and easterly at night (Brandi et al., 2024). The red circle marks the AQS Phoenix JLG Supersite, representing the central air quality monitoring location. In addition, red curves delineate the $O_3$ nonattainment area, which is of regulatory importance due to specific air quality management requirements.

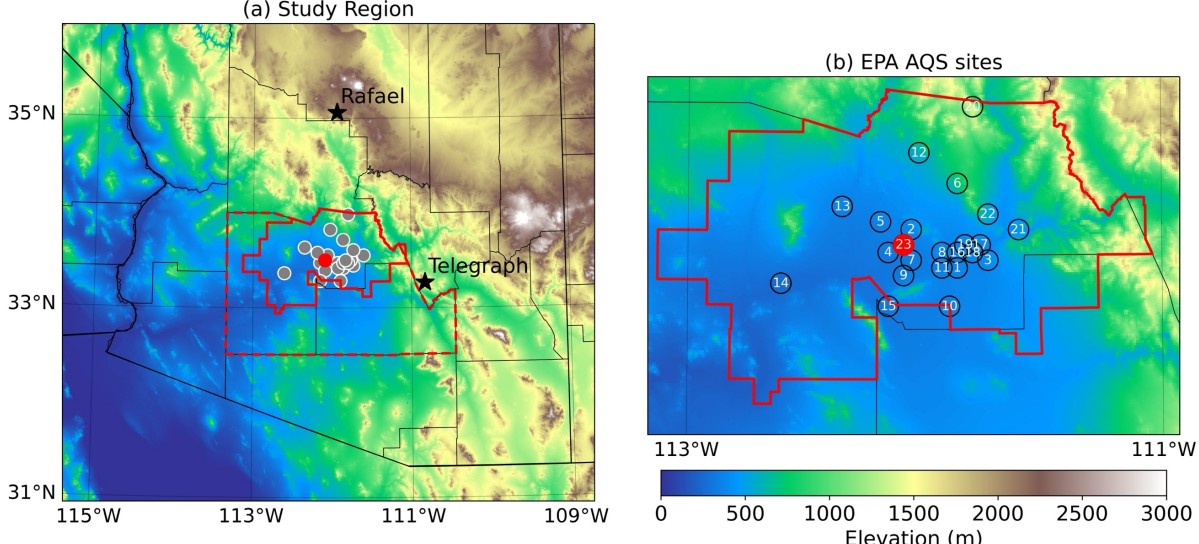

**Figure 1. (a) Topographic map of the Phoenix Metropolitan Statistical Area (indicated by red dashed lines) in Arizona, showing USEPA/AQS monitoring sites as filled circles. The red circle highlights the Phoenix JLG Supersite. The $O_3$ nonattainment area is outlined with a solid red border. Stars mark the locations of the two largest wildfires in June 2021, the Rafael and Telegraph fires. (b) A closer view of EPA AQS sites within the nonattainment area, with sites numbered and positioned according to their geographic locations. The JLG Supersite is designated as AQS site number 23.**

Throughout this study, "Phoenix" refers specifically to this nonattainment area, highlighting its relevance to targeted air quality strategies. The locations of the Rafael and Telegraph fires are marked with stars. The Telegraph Fire is located to the southeast of Phoenix and Rafael Fire is located to the north of Phoenix. Panel (b) of the figure illustrates the distribution of AQS

monitoring sites within the nonattainment area. Each site is numbered and geolocated, offering an observational network for tracking $O_3$ and other air pollutants. Lists of the site locations and associated names are provided in Supplement Table S1. The spatial arrangement of monitoring sites facilitates a spatiotemporal analysis and assessment of pollutant level enhancements from wildfire smoke and other sources of emissions.

## 2.2 WRF-Chem setup

The Weather Research Forecasting with Chemistry (WRF-Chem) (Grell et al., 2005) model v4.4 is utilized here to simulate wildfire activities and study tropospheric $O_3$ pollution. Meteorological initial and lateral boundary conditions are supplied every six hours by the Global Forecast System (GFS) with a horizontal grid spacing of 1° and 12-km NAM (North American Mesoscale Forecast System), while chemical initial and boundary conditions are provided by the Whole Atmosphere Community Climate Model (WACCM) for chemistry (Marsh et al., 2013; Tilmes et al., 2015). Biogenic emissions are calculated online using the Model of Emissions of Gases and Aerosols from Nature (MEGAN, version 2.1) (Guenther, 2007; Guenther et al., 2006), based on the simulated meteorological conditions during the WRF-Chem runs. The anthropogenic emissions used in this study are obtained from 2017 National Emissions Inventories (NEI2017) data provided by the US EPA (https://www.epa.gov/air-emissions-inventories/2017-national-emissions-inventory-nei-data) with a 4 km grid spacing covering the U.S. and surrounding land areas. Biomass burning emissions are calculated using the Fire Inventory from NCAR (FINNv2.5) (Wiedinmyer et al., 2023) and the online plume-rise model (Freitas et al., 2007). FINNv2.5 is based on fire counts derived from Moderate Resolution Imaging Spectroradiometer (MODIS) and Visible Infrared Imaging Radiometer Suite (VIIRS) active fire detection (Wiedinmyer et al., 2023). A summary of the model configuration, parameterization, and a comprehensive model evaluation against multiple observational and reanalysis datasets are provided in Guo et al. (2024). Note that the WRF-Chem model used in this study is coupled with the radiative effects of aerosols, such as smoke, on atmospheric temperature and photochemistry. Both direct and indirect effects of aerosols were turned on in our simulations. These effects are accounted for through radiative transfer calculations, incorporating aerosol optical properties like absorption and scattering. As a result, heavy smoke can reduce the amount of solar radiation reaching the surface, leading to surface and boundary-layer cooling. Furthermore, the attenuation of ultraviolet (UV) radiation by smoke can suppress photochemical $O_3$ production, influencing the atmospheric chemical

environment. Our simulation period focuses on June 2021, targeting multiple wildfire activities near Phoenix as described in section 2.1.

**2.3 $O_3$ tags and experiment design**

To better understand the impacts of wildfire emissions on urban environmental settings, a species tagging technique was employed within the WRF-Chem model following recent demonstrations (Emmons et al., 2012; Gao et al. 2016; Lupaşcu and Butler, 2019; Butler et al., 2018; Butler et al., 2020). Emmons et al. (2012) first introduced a method for tracking the sources of $O_3$ in the troposphere using a tagging approach within various chemical transport models, specifically MOZART-4. This tagging mechanism allows for a detailed attribution of $O_3$ to its precursor emissions sources, providing insights into how different sources contribute to overall $O_3$ levels. Later on, Butler et al. (2018) applied the tagging mechanism for tracking the sources of tropospheric $O_3$ within the Community Earth System Model (CESM) version 1.2.2 and presented an updated version with comparison to Emmons et al. (2012). Lupaşcu and Butler (2019) then implemented the tagging mechanism within the WRF-Chem model to explore the origins of surface $O_3$ across Europe by distinguishing the contributions of different $NO_x$ emission sources to $O_3$ concentrations in various European regions.

Following Lupaşcu and Butler (2019), here we apply the tagging technique in the WRF-Chem model to quantify the contributions of different $NO_x$ sources by not just tagging different regions but also different types of emissions. Our tags include four main categories: 1) regions that are local and adjacent, such as Arizona, California, and Mexico; 2) emission types, including anthropogenic sources and fires; 3) tracers, including NO, $NO_2$, CO, and reaction products like $O_3$, O, and the corresponding $NO_y$ reservoir species; and 4) background $O_3$ from initial and boundary $O_3$ levels. Note that these tracers undergo the same processes (advection, mixing, convection, chemical loss, deposition) within the continuity equation associated for each species in the model but they do not interact and affect changes in the modeled chemical system.

To implement the tagging technique in WRF-Chem, several steps must be completed before running the model. First, a tagged gas-phase chemical mechanism is created to incorporate tagged tracers and reactions representing these tagged species, as well as the production and loss of $O_3$ to account for the tagged $NO_x$ emissions. The tagged $O_3$ in the model is represented as tracers that track its production from $NO_x$, as well as its subsequent transport and loss processes. Here, we

assume that $O_3$ peaks in urban areas of Arizona during this study period (June) is under a $NO_X$-limited chemical regime based on our previous studies (e.g., Guo et al., 2024; Greenslade et al., 2024).This new tagged mechanism is modified from the source code of the original MOZART mechanism within the WRF-Chem model. Next, both the anthropogenic and fire emission input files are modified to include tags related to different regions. For each regional tag, such as Arizona, $NO_x$ and CO concentrations from outside Arizona are set to zero. Finally, the tags are initialized, and their boundary conditions are determined by the WACCM model output. The advantage of using WACCM is that it provides tagged CO tracers, including global biomass burning, North American anthropogenic emissions, and continental transport from regions such as East Asia, Europe, and Africa.

Since meteorological conditions, particularly wind speed and direction, have a significant impact on wildfire activities and plume coverage, we also apply the higher resolution 12-km NAM (North American Mesoscale Forecast System) dataset as the initial and boundary conditions. Evaluations are conducted for each selected case, comparing them against two boundary conditions. The simulations featuring winds and smoke plumes that best match satellite observations are selected. To help evaluate the contribution of wildfire emissions to $O_3$ levels, another set of simulations is performed by removing fire emissions. This serves as a sensitivity test for evaluating the model results with tags.

**2.4 EPA AQS surface observations**

To evaluate the accuracy of our model simulations, we use the surface observations from the Environmental Protection Agency's Air Quality System (EPA AQS). The AQS provides comprehensive air quality data from monitoring stations across the U.S., offering measurements of various pollutants, including $O_3$, particulate matter ($PM_{2.5}$ and $PM_{10}$), and other criteria pollutants. The hourly and daily surface in situ observations of $O_3$ (including MDA8), CO, $NO_2$, and meteorological fields such as temperature, relative humidity, and winds from the EPA AQS monitoring network are used in this study (Demerjian, 2000). A total of 23 sites within the nonattainment area were selected based on their availability of $O_3$ measurements during the study periods, as shown in Figure 1b. The dataset undergoes quality control procedures to filter out any erroneous or incomplete records, ensuring that only high-quality observations are used in our evaluation.

## 2.5 HMS smoke products

The Hazard Mapping System (HMS) smoke products provide detailed daily maps showing the geographic extent and concentration of smoke plumes across the U.S. and surrounding regions. The system integrates various satellite data sources, including the MODIS and VIIRS sensors, GOES (Geostationary Operational Environmental Satellite) imagery, to detect fire locations and estimate smoke coverage. The HMS smoke analysis has been a useful tool in monitoring wildfire

impacts, supporting meteorological forecasting, and informing public safety measures related to air quality (e.g., Brey et al., 2018; Rolph et al., 2009).

The smoke products typically include three types of shapefiles: light, medium, and heavy (NOAA, 2023). Each category includes one or more shapefiles representing the smoke coverage estimated from satellite observations or images. These smoke products are used in this study to identify and

325 select cases when Phoenix is defined as experiencing heavy smoke days.

## 2.6 TROPOMI satellite retrievals

The TROPOspheric Monitoring Instrument (TROPOMI) is a state-of-the-art satellite sensor onboard the European Space Agency's (ESA) Sentinel-5 Precursor (S5P) satellite, launched in October 2017. TROPOMI actively measures tropospheric columnar atmospheric constituents

including $O_3$, CO, $NO_2$, formaldehyde (HCHO). The TROPOMI dataset over Arizona has a spatial resolution of approximately $5.5 \times 3.5$ km² at nadir and provides daily data with an early afternoon (~12-14 PM) overpass time (Ludewig et al., 2020; Van Geffen et al., 2020). The data utilized in this research underwent a quality control process, where a quality assurance value (qa_value) greater than 0.50 was applied for HCHO and CO and a qa_value greater than 0.75 was applied for

$NO_2$. The quality-controlled datasets were then gridded to a resolution of $0.07° \times 0.07°$ for spatial analysis. For days with a lack of good quality data over the study domain, the data were further re-gridded to a coarser spacing of $0.2° \times 0.2°$ to better capture the general spatial pattern of $NO_2$ and HCHO tropospheric columns. TROPOMI $O_3$ data was not used in this study due to limitations in its applicability to our research domain. The high-resolution TROPOMI $O_3$ product primarily

represents total column $O_3$, which is strongly influenced by stratospheric $O_3$ rather than tropospheric levels (Copernicus Sentinel data processed by ESA et al., 2020a). The tropospheric column $O_3$ product is only available for latitudes between -20° and 20°, as it relies on the Convective Cloud Differential (CCD) method, which is most effective in regions with frequent

high convective clouds (Copernicus Sentinel data processed by ESA et al., 2020b; Heue et al., 2021). This limitation excludes our study area. Additionally, the CCD method has shown stronger utility in tropical regions (Cazorla & Herrera, 2022). While TROPOMI $O_3$ profile data is available, calculating tropospheric columns requires additional processing and extensive validation, which is beyond the scope of this study. Furthermore, $O_3$ profiles do not directly represent surface $O_3$ levels, unlike $NO_2$, which has a shorter lifetime and is primarily associated with surface emissions from combustion sources.

## 3. Results and discussions

This section is divided into two main parts. Section 3.1 provides an overview of the fire events, including both observed and simulated air quality conditions at the surface and throughout the tropospheric column, highlighting the vertical extent of smoke plumes and how well they are represented in the model. Section 3.2 focuses on source attribution using WRF-Chem tagging, further broken down into two key components: overall contributions to monthly $O_3$ and CO extremes (Section 3.2.1) and a detailed investigation of two major smoke events. his section explores fire-related contributions $O_3$ levels in Phoenix, the influence of wildfires on atmospheric chemistry and meteorology, and a comparative assessment of the chemical regime driving $O_3$ production, particularly through the formaldehyde-to-nitrogen dioxide ratio (FNR). To evaluate wildfire impacts, we compare FNR simulations in WRF-Chem (with and without fire emissions) against TROPOMI satellite-derived FNRs, providing insights into how fire emissions modify $O_3$ chemistry.

### 3.1 Air quality (AQ) setting during June 2021 fire events

Throughout June 2021, Phoenix experienced an intensified heatwave and drought conditions conducive for wildfire activity. Figure 2 presents the observed daily variations of pollutant levels for June 2021 from the EPA AQS at the Phoenix JLG Supersite compared to results from the TROPOMI satellite. Data on temperature (T) shows the heatwave began on June 12, with the daily maximum temperature reaching 43°C and remaining at least that high until June 20. The month of June represent a dry summer in Arizona, which is characterized by intense heat and arid conditions, prior to the onset of the North American Monsoon in early July where rainfall typically starts (monsoon summer). The Maximum Daily 8-Hour Average Ozone (MDA8 $O_3$) levels were around

50-70 ppb until June 10, when $O_3$ began to increase, exceeding the EPA National Ambient Air Quality Standard (NAAQS) standard (70 ppb) along with elevated surface concentrations of CO and $NO_2$, and T. A notably high MDA8 $O_3$ level (100 ppb) was observed on June 15. MDA8 $O_3$ then decreased to below NAAQS levels on June 17 up until June 27. Surface CO levels generally followed the $O_3$ variation, ranging between 400-700 ppb.

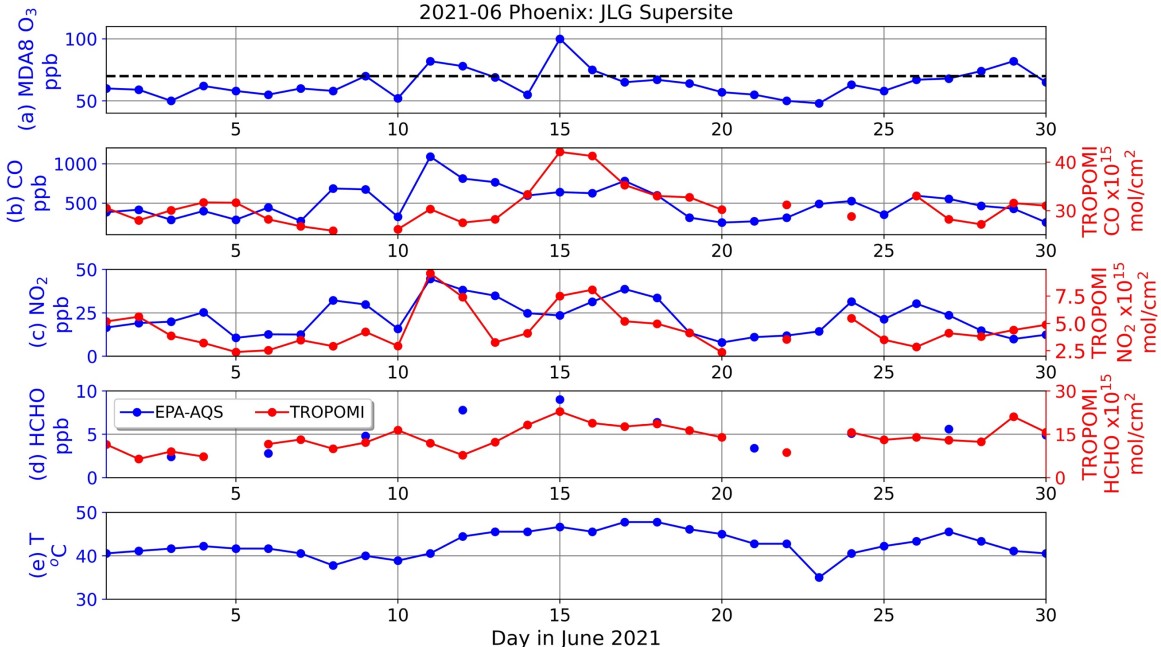

**Figure 2. Observational daily variations of surface (a) MDA8 $O_3$, (b) CO, (c) $NO_2$, (d) HCHO, and (e) temperature (T) from EPA AQS at Phoenix JLG supersite (blue), as well as column (b) CO, (c) $NO_2$, and (d) HCHO concentrations from TROPOMI satellite (red) in June 2021. The NAAQS 2015 standard is denoted as black dash line. Note that the daily AQS CO, $NO_2$, and T values shown represent the daily maximum while the HCHO value is the daily mean.**

However, a noticeable peak in surface CO, exceeding 1000 ppb, was observed on June 11, which was not shown in the TROPOMI tropospheric column CO, while both surface and tropospheric column $NO_2$ levels exhibited a significant peak, indicating that emissions were mostly within the planetary boundary layer. Conversely, the discrepancies between $NO_2$ surface and column levels beginning June 13 suggest different sources for surface and tropospheric $NO_2$ and CO emissions, particularly on June 15, when MDA8 $O_3$ exceeded 100 ppb; surface $NO_2$ and CO was relatively low, while column $NO_2$ and CO were high. The peak period beginning June 14 of HCHO concentration was also captured by both AQS and TROPOMI observations. In Guo et al. (2024),

they showed that during the period with elevated temperature (June 12-20), relative humidity is low but normal for June in Arizona. The high temperature resulted in an increase of isoprene and HCHO simultaneously.

As mentioned earlier, the Telegraph Fire began on June 3 and lasted for one month, while the Rafael Fire started on June 18. Guo et al. (2024) showed that in the month of June, the prevailing wind over Phoenix was mostly southwesterly, limiting the impacts of these wildfires on Phoenix to certain days when winds shifted direction and brought smoke plumes to the city. After reviewing the HMS smoke data for June 2021, we identified two smoky periods that might have potentially

influenced surface $O_3$ concentrations over Phoenix.

The first selected case is on 15 June 2021. On this day, multiple sites within the nonattainment area observed $O_3$ exceedances (>70 ppb). An excessive heat warning was issued and remained in effect through the end of the week, with temperatures 10 to 15 degrees above average (Gard and Garrett, 2021). The wind shifted from southwesterly to northeasterly, bringing the Telegraph Fire

plumes to Phoenix. The second case is on 26 June 2021, when smoke from the Rafael Fire spread to the north of Phoenix with a change of wind direction from southwesterly to northerly.

Shown in Figure 3 is the heavy smoke coverage from the HMS smoke products over the Phoenix area during two selected cases in June 2021, highlighting the impact of the Rafael and Telegraph fires. In Case I, on 14 June 2021, at 16:00 LT, smoke from both the active Telegraph (southeast of

Phoenix) Fire spread primarily to the northeast. By 15 June 2021, at 16:20 LT, the smoke coverage had expanded significantly, with a dense plume covering central Arizona, including Phoenix. On 16 June 2021, at 11:00 LT, the dense and widespread smoke continued to affect the periphery of Phoenix. In Case II, on 25 June 2021, at 07:00 LT, smoke primarily from the Rafael Fire extends to the east, far away from Phoenix. By 26 June 2021, at 17:00 LT, the smoke plume from Rafael

Fire changed direction to the south and covered the north of Phoenix. Active wildfires contributing to the smoke over Phoenix are marked with yellow stars, indicating the origin and spread direction of the smoke plumes. These two cases are selected for further modeling studies to help understand how near-range wildfires affect the Phoenix metropolitan area.

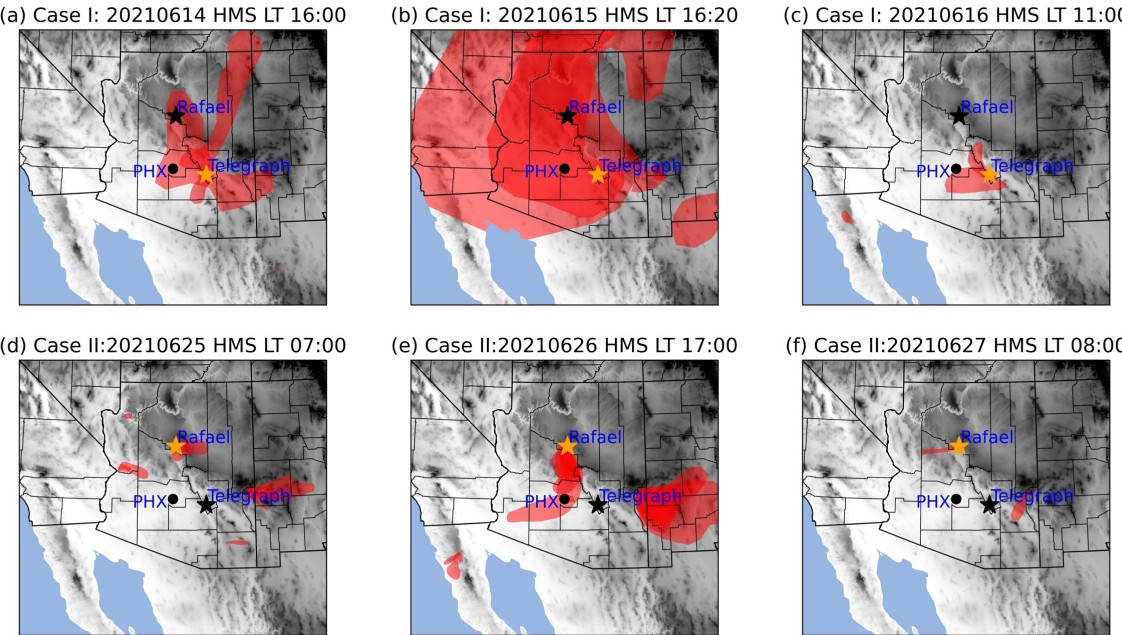

**Figure 3. (a-c) Heavy smoke coverage from Hazard Mapping System (HMS) smoke products for Case I, and (d-f) for Case II over Phoenix area. The active wildfire that was accountable for the smoke over Phoenix is marked as yellow star.**

In Figure S1 we present a series of screenshots from the MODIS Terra Corrected Reflectance map, overlaid with MODIS fires and thermal anomaly products, to depict wildfire activities in Arizona for the above two cases. Similar to Figure 3, the top panel illustrates Case I, focusing on the Telegraph Fire from June 14 to 16. The bottom panel captures Case II, highlighting the Rafael Fire from June 25 to 27. Both cases show visible smoke plumes and thermal anomalies (orange color) indicating active fire regions, with the fire spreading and producing significant amounts of smoke passing Phoenix.

In addition to HMS smoke products, we show in Figure 4 the daily TROPOMI tropospheric columns of HCHO, $NO_2$ and CO during smoke periods for Case I over the Phoenix area. In Case I, the HCHO levels initially show low levels, with scattered low concentrations on June 13 except the east of Telegraph Fire. By June 14, there is a significant increase in HCHO, especially northeast of Phoenix, correlating with the smoke plume from the Telegraph Fire, as seen in Figure 3a. On June 15, the elevated HCHO levels were more dispersed, affecting mainly the south of Phoenix. By June 16, HCHO tropospheric column decreased to the normal levels over Phoenix. For $NO_2$, June 13 shows low levels with a typical urban anthropogenic emissions spatial profile. June 14

exhibits a significant increase in NO₂, particularly northeast of Phoenix, similar to the HCHO
distribution. On June 15, NO₂ levels are high over a wider area, including Phoenix and the path of
plumes (Figure 3b). By June 16, NO₂ levels decrease but remain elevated.

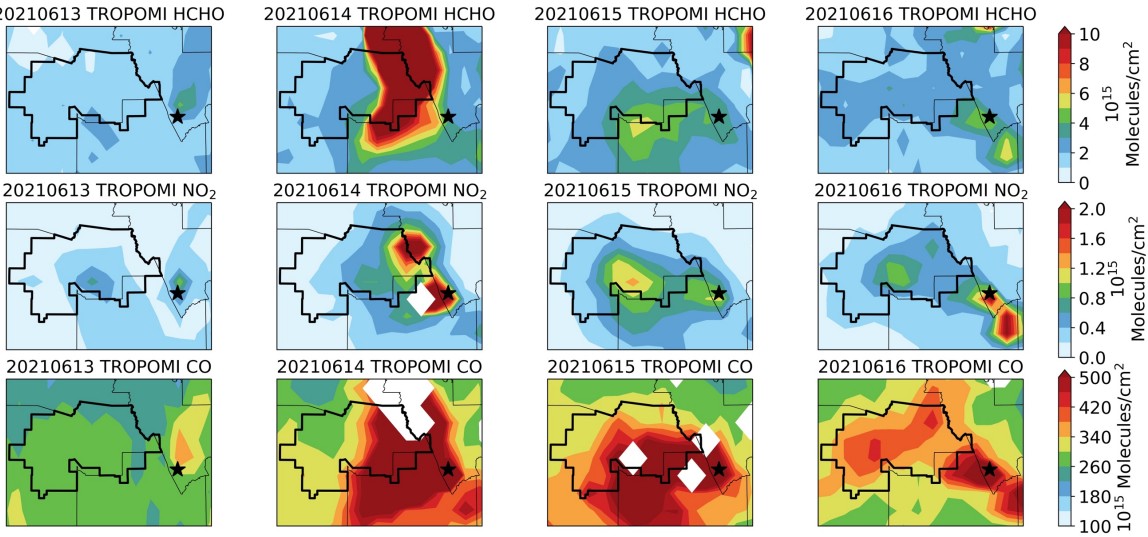

**Figure 4. TROPOMI tropospheric columnar HCHO (top), NO₂ (middle), and CO (bottom)
during the smoke periods for Case I over Phoenix area. The black polygon lines represent the
EPA designated Phoenix-Mesa nonattainment area. The grid resolution is 0.2°. Grids without
available data are marked as white space.**

A similar pattern has been observed in CO, where its high concentrations are closely correlated
with HCHO, NO₂, and smoke coverage, as shown in Figure 3. The TROPOMI results for Case II
are presented in Figure S2.

Similar to the TROPOMI satellite observations, we also examined the model simulation results.
Figure 5 presents the WRF-Chem simulated tropospheric columnar values of HCHO, NO₂, and
CO at 14:00 local time (same time as the TROPOMI observations) during the smoke periods for
Case I. Comparing Figures 4 and 5, on June 13 as a pre-smoke day, the HCHO levels in the city
region are comparable, with values primarily below $6 \times 10^{15}$ molecules/cm², while the Telegraph
Fire burning area reached over $10 \times 10^{15}$ molecules/cm². This day represents a typical distribution
of urban pollutants over Phoenix. By June 14, levels of HCHO, NO₂, and CO increased
significantly, particularly in the southeastern part of Phoenix, although the magnitude and spatial
patterns appear to differ from the satellite observations in Figure 4. On June 15, the tropospheric
columnar values decreased but the wildfire signal remained significant until June 16, when the

spatial pattern returned to typical conditions. Additional WRF-Chem simulated results for Case II are available in Figure S3-S4.

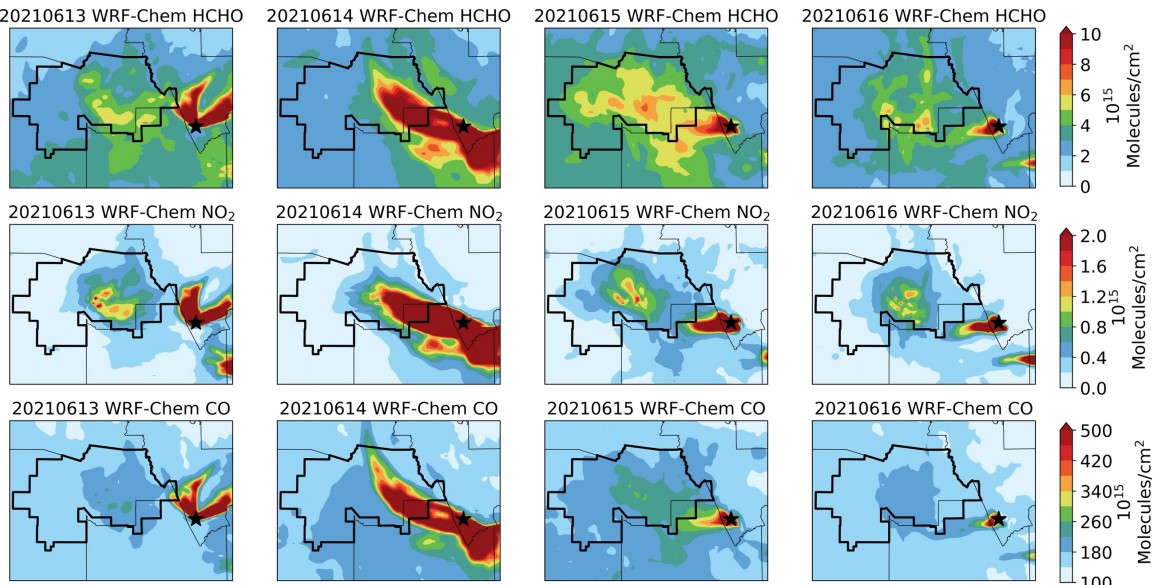

**Figure 5. WRF-Chem simulated HCHO, NO₂, and CO tropospheric columns at local time 14:00 during the smoke periods for Case I over Phoenix area.**

In summary, observations from HMS, TROPOMI, and WRF-Chem models indicate that the Telegraph Fire had a significant impact on Phoenix air quality during June 14-15. While the rise of plumes during wildfires greatly influences the columnar concentrations of pollutants by transporting smoke and emissions higher into the atmosphere, the mixing levels within the surface or planetary boundary layer are more important to the overall air quality and pollutant distribution

as the more immediate impact on public health is expected at ground-level.

We show in Figure 6 the WRF-Chem simulated surface concentrations of HCHO, NO₂, and CO. Since each case involves two sets of simulations using GFS and NAM meteorological boundary conditions, the selection of the model results is based on an initial evaluation against AQS and satellite observations. For Case I, the results from the GFS simulations demonstrate better

agreement, while for Case II, the NAM simulations show better alignment of smoke. Comparing these with the columnar levels of NO₂ and CO in Figure 5, it is evident that the extent of the smoky day on June 14, as observed from HMS and TROPOMI, is not reflected at the surface level, whereas the smoky day on June 15 is apparent in both surface and columnar concentrations. Additionally, for HCHO, increases are also observed at the surface on June 14. This discrepancy

indicates that on June 14, the wildfire smoke was primarily affecting atmospheric layers aloft without significantly impacting the ground level, while on June 15, the smoke was more distributed in the lowermost troposphere, increasing the surface pollution concentrations. Model results of tropospheric column and surface HCHO, $NO_2$, and CO for Case II are provided for reference in the Supplement as Figures S1-S2, respectively.

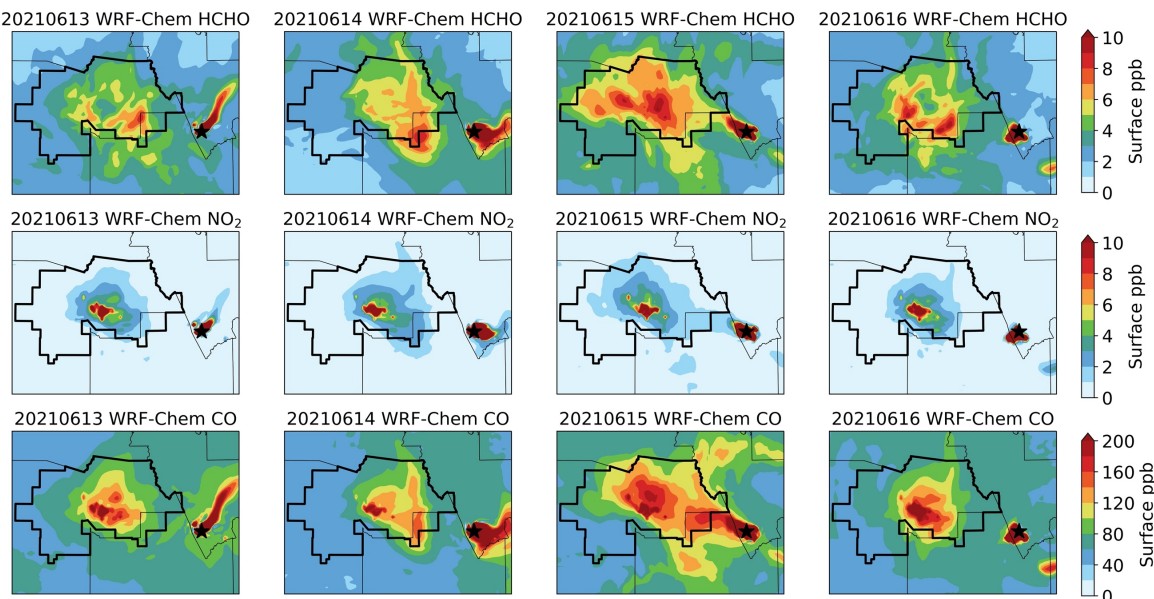

**Figure 6. Same as Figure 5, but for surface concentrations.**

## 3.2 Source attribution with tags

An extensive evaluation of the same configuration of the WRF-Chem model using the MOZART chemical mechanism, except for the tags, has been presented previously by Guo et al. (2024). Briefly, our evaluation showed a Pearson correlation coefficient (R) of 0.81 for modeled and observed $O_3$ over Phoenix with a mean bias (MB) of -2.9 ppb and 1.0 ppb for hourly and MDA8 $O_3$, respectively. For CO and $NO_2$, the normalized bias is 7.1% and 5.3%, respectively. The model simulations also show that surface formaldehyde-to-nitrogen dioxide ratio (FNR), which is an indicator of chemical regime affecting $O_3$ production varies from a VOC-limited regime in the most populated areas to a transition between VOC-limited and NOx-limited regimes throughout the metro area. For the FNR threshold, we adopt the same approach as Guo et al. (2024), following the methodology of Duncan et al. (2010), who linked the FNR with surface $O_3$ in model simulations. According to this framework, the sensitivity regime is defined as follows: when FNR is less than 1, it is classified as VOC-limited; values between 1 and 2 indicate a transitional regime;

and an FNR greater than 2 indicates a $NO_x$-limited regime. Here in this study, our discussion of the model results is focused on the month of June 2021, a period marked by active wildfires over Arizona against a backdrop of not only an $O_3$ chemical regime that is in transition to $NO_X$-limited but also of drought and heat wave conditions.

We first provide an analysis of the contribution of different source regions and emission types to the monthly CO and MDA8 $O_3$ concentrations to understand the overall pollution sources in the State of Arizona. Then, we focus on the analysis of $O_3$ during smoky days by examining the two selected cases described in Section 3.1. Note that in this study, "background $O_3$" and "background CO" refer to the residual concentrations after subtracting contributions from tagged anthropogenic and fire emissions. For both $O_3$ and CO, this background includes contributions from natural sources, such as biogenic emissions (e.g., isoprene for $O_3$ and CO), soil and lightning $NO_X$ for $O_3$, and stratospheric ozone, as well as long-range transport from both natural and anthropogenic sources. During heatwave events, background $O_3$ and CO can be particularly elevated due to enhanced biogenic emissions and other natural fluxes. Thus, the background levels of $O_3$ and CO in this context represent a combination of regional and global influences from natural sources and transported components, not solely remote anthropogenic contributions.

### 3.2.1 Monthly CO and $O_3$ extremes for June 2021

Shown in Figure 7 is an overview of CO concentrations in Arizona during June 2021, highlighting the impact of various sources on CO distribution. Each panel represents the 90th percentile for the entire month of different CO and its sources: (a) total CO, (b) background CO levels, (c) anthropogenic CO sources, (d) CO from California anthropogenic sources, (e) CO from Arizona anthropogenic sources, (f) CO from Mexico anthropogenic sources, (g) CO from Arizona wildfires, and (h) CO from Mexico wildfires.

Comparing the total CO concentrations (Figure 7a) with anthropogenic CO (Figure 7c), we can see a clear signature of anthropogenic activities in cities such as Phoenix (PHX), Tucson (TUS), and Las Vegas, located in the upper left corner of the map. The "background" CO levels (Figure 7b) are generally constant, ranging between 50 and 70 ppb across the region, which is closely related to international or long-range transport as well as global secondary CO formation. When examining anthropogenic sources, contributions are tagged separately for California (Figure 7d), Arizona (Figure 7e), and Mexico (Figure 7f).

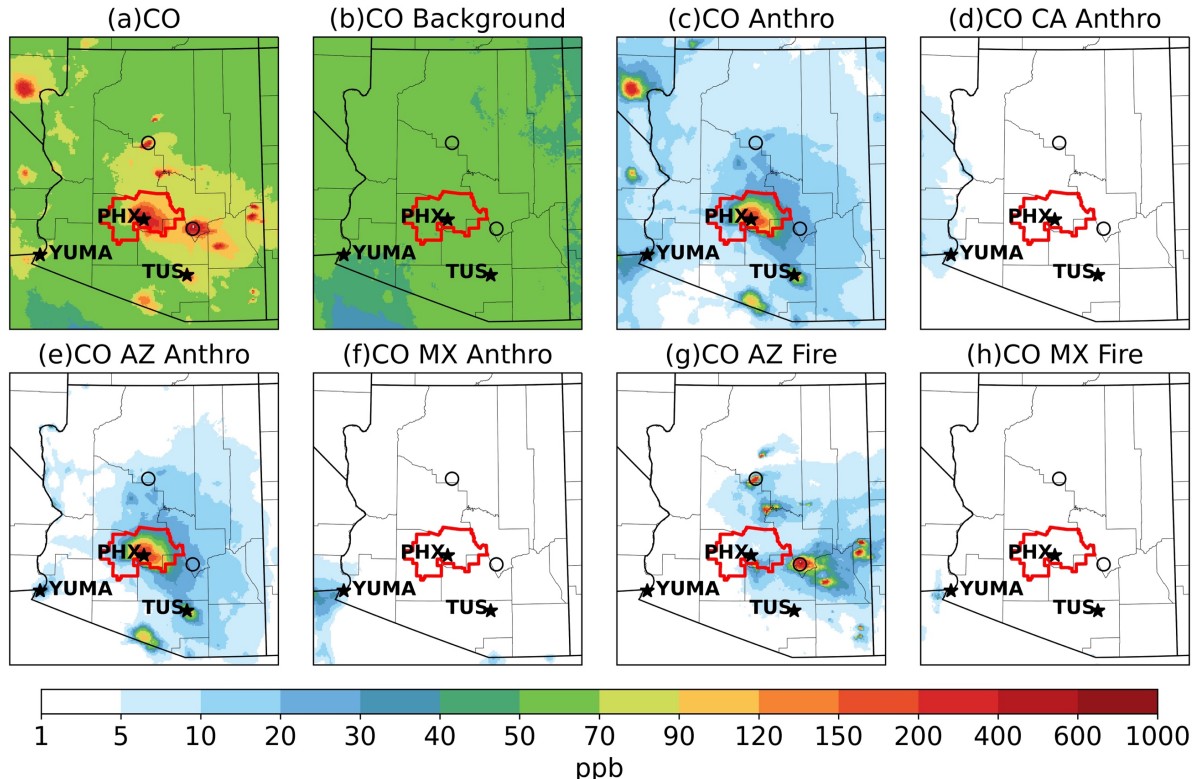

**Figure 7. WRF-Chem simulated 90<sup>th</sup> percentile of surface CO concentrations during June 2021 for total CO (a), and the contributions from different CO sources (b-h). Each panel represents different aspects of CO: (a) total CO, (b) background CO, (c) anthropogenic CO sources, (d) CO from California anthropogenic sources, (e) CO from Arizona anthropogenic sources, (f) CO from Mexico anthropogenic sources, (g) CO from Arizona wildfires, and (h) CO from Mexico wildfires. Key locations such as Phoenix (PHX), Tucson (TUS), and Yuma are marked as stars on the maps. Telegraph and Rafael fires are denoted as unfilled circles.**

The dominant contributions are seen around Arizona's urban areas, particularly Phoenix and Tucson, highlighting the impact of local urban emissions. CO from Mexico also influences southwestern boundaries with Arizona, particularly the city of Yuma, with an estimate of 30 ppb. Contributions from California (Figure 7d) are limited to the state boundaries, with only minor impacts to surface CO (~5 ppb) during this period. As shown in Figure 7g, wildfires in Arizona

notably elevate CO levels, especially in areas downwind of active fires with six major wildfire activities identified.

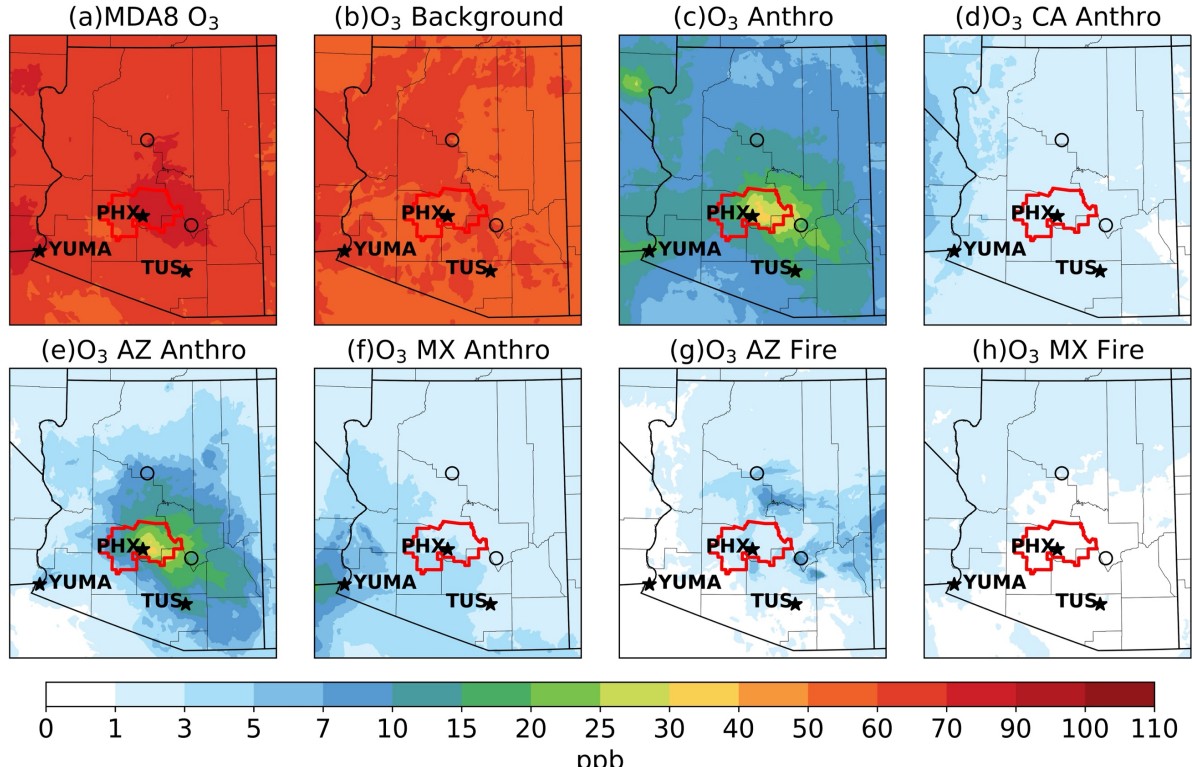

**Figure 8. WRF-Chem simulated 90$^{th}$ percentile of O$_3$ concentrations during June 2021 for (a) MDA8 O$_3$, and contributions from different sources as (b) background O$_3$, (c) O$_3$ from anthropogenic sources, (d) O$_3$ from California anthropogenic sources, (e) O$_3$ from Arizona anthropogenic sources, (f) O$_3$ from Mexico anthropogenic sources, (g) O$_3$ from Arizona wildfires, and (h) O$_3$ from Mexico wildfires. Key locations such as Phoenix (PHX), Tucson (TUS), and Yuma are marked as stars on the maps. Telegraph and Rafael fires are denoted as unfilled circles.**

Additionally, we examined MDA8 O$_3$ using the tags, as presented in Figure 8. Note that Figure 8 represents 90$^{th}$ percentile MDA8 O$_3$ and its corresponding contributions during the month of June rather than instantaneous O$_3$ concentrations. We see that the MDA8 O$_3$ concentrations are predominantly high across the region, with the highest levels observed around Phoenix. Figure 8b indicates that the background O$_3$ levels are uniformly high, approximately 50-60 ppb, suggesting

that even in the absence of local sources, O$_3$ concentrations remain elevated due to regional and global influences on a monthly basis. Figure S8 provides a more detailed spatial distribution of the monthly mean and 90th percentile background O$_3$ estimates from WRF-Chem tagging. The monthly mean background O$_3$ ranges from 45–50 ppb over the Phoenix Metro area to 50–55 ppb in northwestern Arizona, where most areas are rural and have been identified by Greenslade et al.

(2024) as representative of background $O_3$. Notably, observed $O_3$ levels at Grand Canyon and Alamo Lake from 2020–2022 (Greenslade et al., 2024) averaged 63–65 ppb, reinforcing these estimates. The 90th percentile background $O_3$, which reflects extreme values comparable to the $O_3$ design value (ODV), ranges from 50–60 ppb in Phoenix and 60–65 ppb across rural Arizona. These background estimates align with recent studies, including the $69 \pm 2$ ppb reported by Parrish et al.

(2025) based on ODVs across monitoring sites, the 56–66 ppb found by Hosseinpour et al. (2024) using multivariate regression and machine learning to adjust CAMx simulations, and the 60–70 ppb reported by Jaffe et al. (2018) as the 4th highest North American Background (NAB) MDA8 $O_3$ value at rural locations using the GFDL AM3 model.

We show in Figures 8c to 8f a regional decomposition of the anthropogenic contributions to $O_3$

levels. Figure 8c represents all anthropogenic sources, revealing significant contributions, especially around urban centers like Phoenix and Tucson. Figure 8d shows the small impact of California's anthropogenic emissions on Arizona's $O_3$ levels during this period only reaching ~3 ppb in Yuma. In contrast, Arizona's anthropogenic contributions to Arizona's $O_3$ levels (Figure 8e) are substantial (as expected), ranging from 25 to 30 ppb within the nonattainment area. Mexico's

anthropogenic contributions (Figure 8f) have a larger impact to $O_3$ than they do for CO (Figure 7f) in terms of spatial coverage, affecting most of the southern Arizona regions and even reaching Phoenix at 3 ppb. The magnitude is also higher, reaching 10 ppb for Yuma.

Similar to CO, Figures 8g and 8h focus on $O_3$ contributions from wildfires in Arizona and Mexico, respectively. However, while CO is directly emitted from wildfires, $O_3$ is chemically formed from

precursors such as VOCs and $NO_x$ transported with the smoke. Consequently, the patterns of $O_3$ differ from those of CO. $O_3$ can have a larger impact due to the transport of these precursors, leading to significant $O_3$ formation even far from the wildfire sources. Figure 8g shows that wildfires in Arizona contribute notably to $O_3$ levels, particularly in areas close to and downwind of the fires. $O_3$ concentrations range from 1 to 10 ppb, with the highest levels observed near the

wildfire locations. The influence of these wildfires extends towards the east and southeast, consistent with the prevailing winds being eastward, and indicating the transport of $O_3$ precursors and subsequent formation of $O_3$ in these areas.

Figure 8h highlights the influence of wildfires in Mexico on $O_3$ levels in Arizona, particularly affecting the southern and southwestern parts of the state. The contributions from Mexico wildfires

are less than 3 ppb. The transport of smoke and $O_3$ precursors from Mexico affects a broader area than CO, reaching as far as Phoenix and diminishing farther north. This underscores the effect of cross-border wildfire emissions on $O_3$ levels and air quality in southern Arizona, particularly in border regions like Yuma.

We can see in Figures 7 and 8 that Yuma, which is located at the boundaries of Mexico and California, are influenced by local, regional, and transboundary CO and $O_3$. In Figure 9, we present the modeled and observed hourly $O_3$ concentrations at local time from Yuma monitoring site (AQS site number: 04-027-8011) for the period between June 14 and June 19, highlighting the contributions from various sources. Two episodes of hourly surface $O_3$ exceeding 70 ppb are observed on June 15 and June 17, which the WRF-Chem model generally captures, although some
discrepancies exist.

The shaded areas reveal the contributions from different sources: background $O_3$, local and regional anthropogenic emissions, and wildfire emissions from Arizona and Mexico. Figure 9 shows that $O_3$ levels in Yuma are largely dominated by the background level, primarily from long-range transport and natural sources. The exceedances of the NAAQS 70 ppb $O_3$ standard in Yuma
were significantly influenced by a peak in this background contribution on June 15[th] and 17[th] when the background made up ~65% and ~70%, respectively, of the total daytime $O_3$. On June 15[th] and 17[th], the anthropogenic contributions from Arizona were 20% and 10%, respectively, and the anthropogenic contributions from Mexico were 8% and 13% respectively. We note, however, that these are modeled results, and the modeled peaks on June 15[th] and 17[th] are 16 to 30% different
from the measurement peaks, overestimating on June 15[th] and underestimating on June 17[th].

Figures 7-9 demonstrate the complex interplay of local, regional, and transboundary sources in determining CO and $O_3$ levels. By examining the contributions of local anthropogenic emissions, wildfire emissions, and regional influences from neighboring states and countries, as well as background levels, these figures provide new perspectives of air quality in the region.

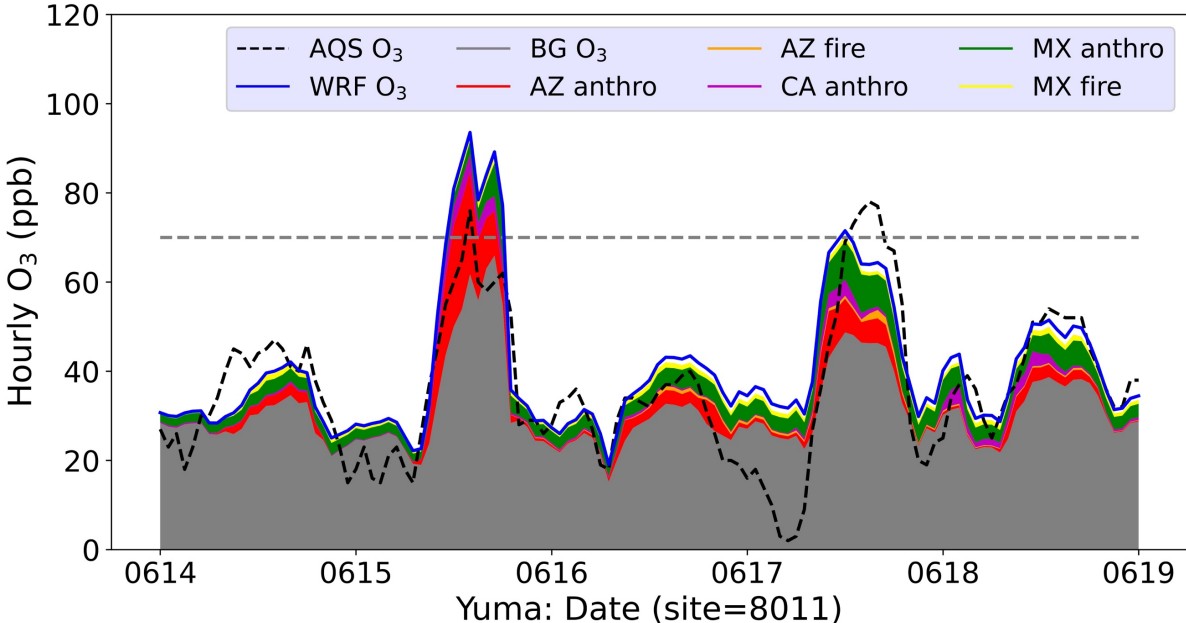

**Figure 9. Hourly O₃ concentrations (in ppb) at the Yuma monitoring site (site number: 04-027-8011) between 14-19 June 2021, at local time. The dashed black line represents observed O₃ levels from the AQS, while the solid blue line shows WRF-Chem simulated O₃ concentrations. Shaded areas indicate contributions from various sources: background O₃ (gray), Arizona wildfires (orange), anthropogenic emissions from Mexico (green), anthropogenic emissions from Arizona (red), anthropogenic emissions from California (purple), and Mexico wildfires (yellow).**

### 3.2.2 Smoky day O₃ analysis

**Fire Contributions to Phoenix O₃.** The detailed analysis presented in Section 3.2.1 provides an overview of the key sources of pollution during a fire season in June. In this section, we examine the impact of wildfire smoke plumes on urban areas by examining two specific smoky days (two cases) with a focus on the Phoenix metropolitan area, where the cases are described in Section 3.1. To assess the effects of fire emissions on O₃ concentrations, we conducted an additional set of WRF-Chem simulations without fire emissions for the same period. The simulations without fire emissions serve as a model sensitivity test to evaluate the impact of wildfires.

Figure 10 illustrates the impact of fire emissions on the MDA8 O₃ concentrations for the two cases. The top panels represent Case I for June 15 (a) without fire emissions and (b) with fire emissions. Similarly, the bottom panels depict Case II for June 26 (c) without fire emissions and (d) with fire

emissions. The comparison between the left and right panels highlights the significant contribution of wildfire emissions to $O_3$ levels in the Phoenix metropolitan area.

In Case I (June 15), the presence of fire emissions (Figure 10b) leads to a substantial increase in MDA8 $O_3$ concentrations, exceeding 110 ppb in areas directly affected by the wildfire plumes. This is in stark contrast to the scenario without fire emissions (panel a), where $O_3$ levels remain below 90 ppb. The path of the elevated MDA8 $O_3$ in Figure 10(b) aligns with the HMS smoke coverage depicted in Figure 3.

For Case II (June 26), a similar pattern is observed, albeit with a much weaker intensity. The inclusion of fire emissions (Figure 10d) also results in elevated MDA8 $O_3$, with peak values reaching around 90 ppb, while without fire emissions (Figure 10c), $O_3$ levels are significantly lower, generally below 70 ppb. The spatial distribution of MDA8 $O_{33}$ also aligns with the mean transport pathway of the wildfire plumes.

The AQS observations, indicated by the colored circles, are generally consistent with the model results when fire emissions are included, demonstrating the model's ability in capturing the impact of wildfire emissions on ground-level $O_3$ concentrations. The mean bias between model without fire emissions and observations is -7.9 ppb for case I and 9.7 ppb for case II. When fire emissions are included, the mean bias is reduced to -1.8 ppb and 2.9 ppb for the two cases, respectively.

Overall, the sensitivity simulation suggests that wildfires exacerbate $O_3$ pollution, especially when fire smoke passes through urban areas when photolysis is high. Additionally, it enables us to evaluate the $O_3$ fire tags. Ideally, the difference in $O_3$ concentrations when fire emissions are excluded should match the $O_3$ fire tags. However, studies have shown that this is not always the case mainly due to non-linearity of $O_3$ chemistry to precursor emissions as well as the spatiotemporal heterogeneity of $O_3$ chemical regime. The differences between attributing source contributions through sensitivity or tagging approaches have been noted by several studies (e.g., Grewe et al., 2010; Grewe, 2013; Kwok et al., 2015, Mertens et al., 2021, Maruhashi et al., 2024). These studies reported that the sensitivity method could potentially induce large errors (factor of 2), which depend on the degree of linearity of the chemical system. To better understand our tagging approach, we show in Figure 11 the WRF-Chem simulated daytime (7:00-19:00 LT) average of $O_3$ concentrations for two different cases: Case I on 15 June 2021 (top panels) and Case II on 26 June 2021 (bottom panels). The left panels display the differences in $O_3$ levels between

scenarios with and without fire emissions. The right panels show the daytime average O₃ concentrations attributed to fire emissions (fire tag).

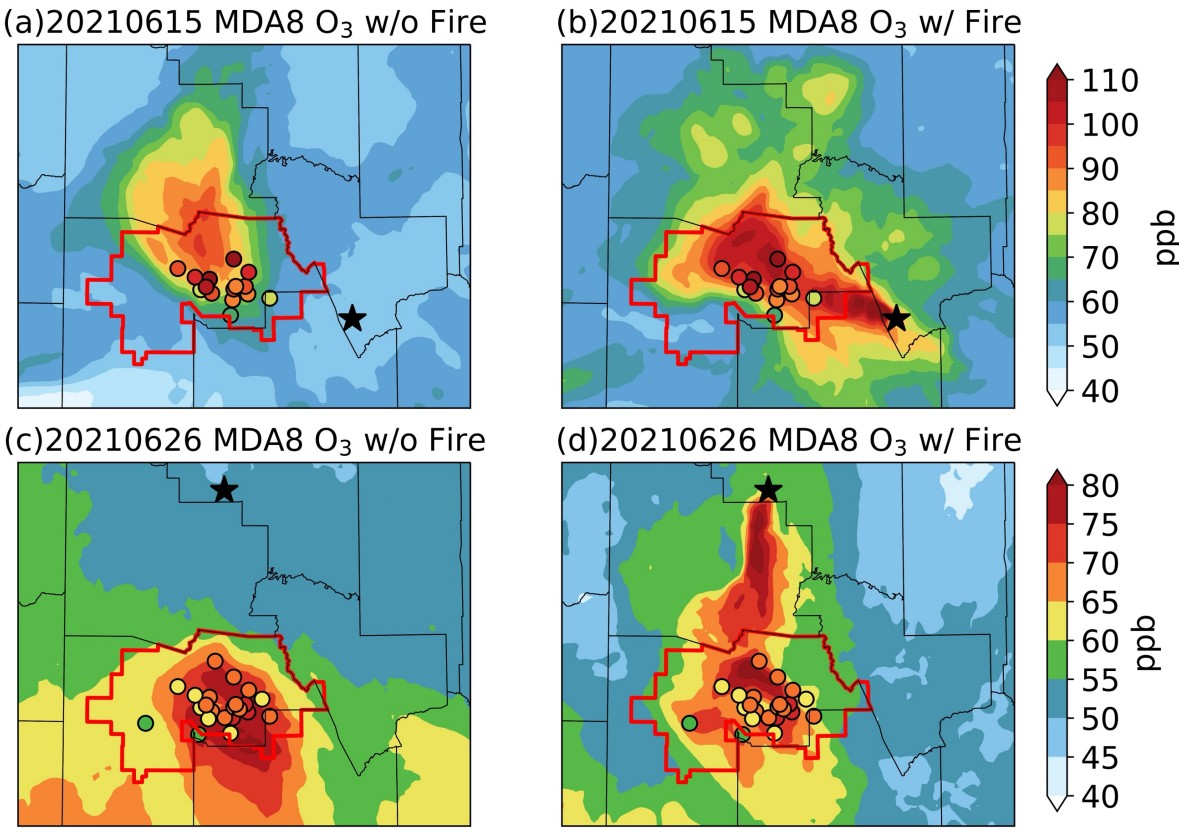

**Figure 10. WRF-Chem simulated MDA8 O₃ concentrations for Case I (15 June 2021, top panels) and Case II (26 June 2021, bottom panels) under two conditions: without fire emissions (left panels) and with fire emissions (right panels). AQS observations are represented by colored circles, excluding sites with missing or low-quality data. Stars indicate the locations of the wildfires (top: Telegraph; bottom: Rafael). The red outline represents the designated nonattainment area.**

The spatial variations observed in the two methods are evidently similar across both cases. However, the values differ by a factor of 1.4, as indicated by the color bar scales, which aligns with previous expectations. Apart from the difference in O₃ magnitude, sensitivity test also shows negative O₃ differences (left panels), which are caused by nonlinear chemical processes. The tagging method does not capture these negative values because the model may not fully represent

the O₃ loss processes, such as O₃ titration or the competition between O₃ production and

destruction pathways. This highlights the importance of combining these approaches in better understanding pollution dynamics.

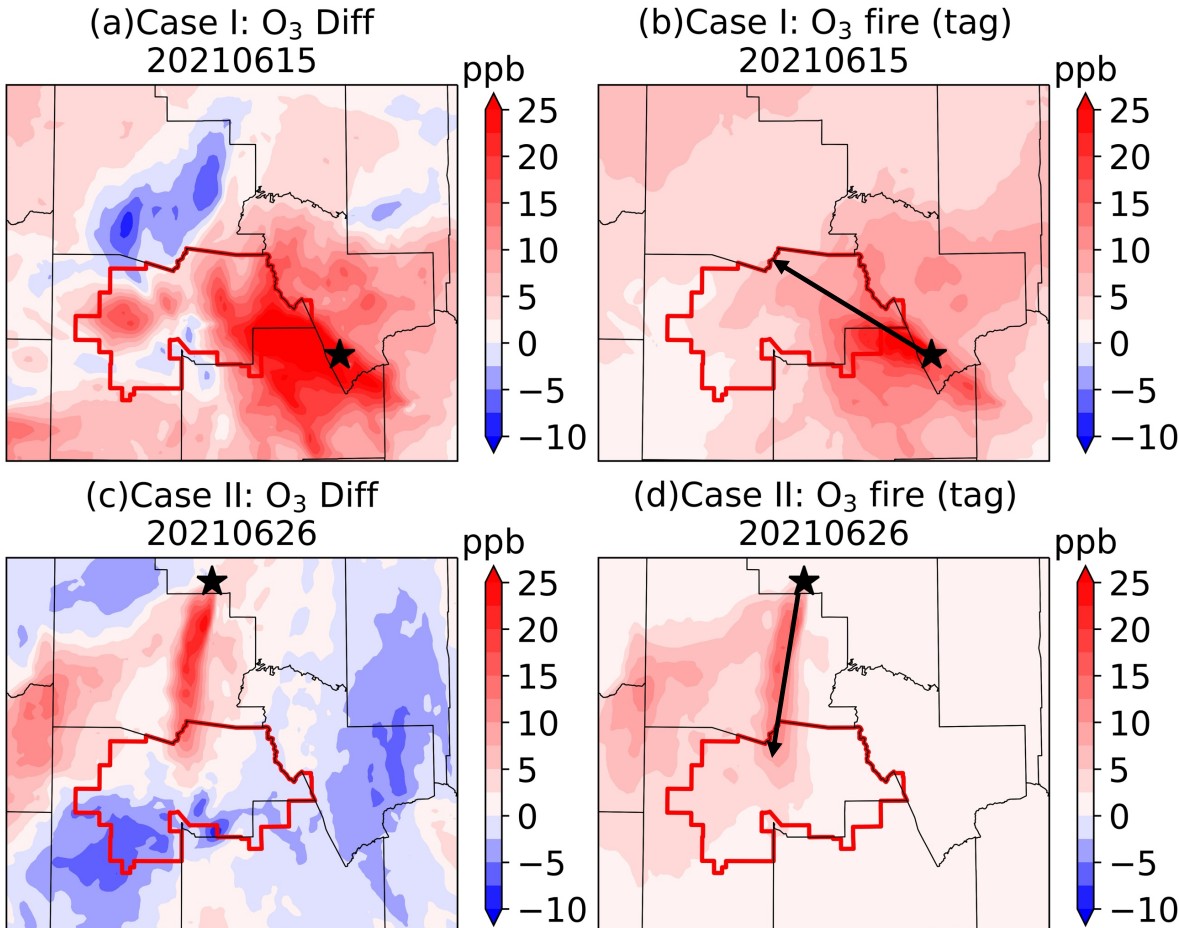

**Figure 11. Daytime (7:00-19:00 LT) average O₃ concentrations simulated by WRF-Chem for Case I (15 June 2021, top panels) and Case II (26 June 2021, bottom panels). The left panels show the difference between scenarios with and without fire emissions, while the right panels depict the daytime average O₃ fire tag. Stars mark the wildfire locations (Telegraph Fire at the top and Rafael Fire at the bottom). The red outline denotes the designated nonattainment area. Note that the color bar scales for left and right panels are different. The black arrows indicate the path of smoke plumes.**

In addition to examining the spatial variations of $O_3$ concentrations, we also present the temporal variations of surface hourly $O_3$ within the Phoenix area in Figure 12, which includes a detailed look at each individual AQS site and the contribution of each $O_3$ tag to the overall $O_3$ levels. First,

a site located under the plume path with significant $O_3$ elevation from smoke is selected for each case. Next, a timestamp is chosen when the $O_3$ fire tag is at its peak to review and compare observations from all AQS sites. The top panels of Figure 12 show the hourly $O_3$ concentrations at AQS sites 7024 and 1010 for Case I and Case II, respectively. The locations and site numbers are detailed in Figure 1 and Table S1. For Case I between June 15 and 17, the peak hourly $O_3$ concentration reached approximately 115 ppb on June 15 at 17:00 local time, aligning with AQS measurements. The contribution from Arizona fire emissions is evident, as indicated by the orange segments in the stacked area chart (Figure 12a). Background $O_3$ levels (gray shading) constitute the largest portion of the total $O_3$, accounting for approximately 50%. Local anthropogenic emissions are the next significant contributor, varying between 24% and 40%, depending on the urban setting of the site. A closer examination of other sites during the $O_3$ peak hour on June 15 reveals that fire contributed $O_3$ is significant across the area, with values around 15 ppb or 15% (Figure 12b). This indicates that the wildfire events during this period had a substantial impact on elevating $O_3$ levels.

For Case II, $O_3$ levels are much lower, peaking at about 80 ppb on June 26 at 11:00 (Figure 12b). Compared to Case I (Figure 12a), the impact of fires on $O_3$ levels is less pronounced. After June 26, $O_3$ levels returned to non-smoky day patterns, with most contributions from local anthropogenic emissions. Figure 12d further illustrates the distribution of $O_3$ sources across multiple sites at 14:00 on June 26, showing fire contributions of 5-10 ppb or approximately 10%. The background $O_3$ levels remain consistent with Case I. The differences between these two cases may be attributed to varying meteorological conditions, fire intensity, and/or the spatial distribution of emissions during the two periods. During Case I, Arizona experienced excessive heat and record high temperatures (Figure 2), and the Telegraph Fire had a larger and longer smoke impact than the Rafael Fire in Case II. Unlike Yuma, as shown in Figure 9, $O_3$ levels in Phoenix are primarily influenced by local emissions, with much smaller contributions from California or Mexico, even with significant contributions from wildfire smoke.

An additional figure comparing the effects of anthropogenic and fire-related emissions on $O_3$ levels for Case I is provided in Figure S5. This figure shows a pronounced diurnal cycle, with $O_3$ levels increasing from early morning, peaking around noon to early afternoon (12 to 1pm), and then declining towards the evening. Our results show significant differences between these two emission sources across three urban settings: suburban, urban, and rural. In the early morning and

early afternoon, $O_3$ levels are predominantly influenced by anthropogenic emissions at most AQS sites. However, in the late afternoon, when a fire smoke plume passed through the Phoenix urban area, the contribution of fire related $O_3$ increases significantly and, in some rural sites, even surpasses local anthropogenic production.

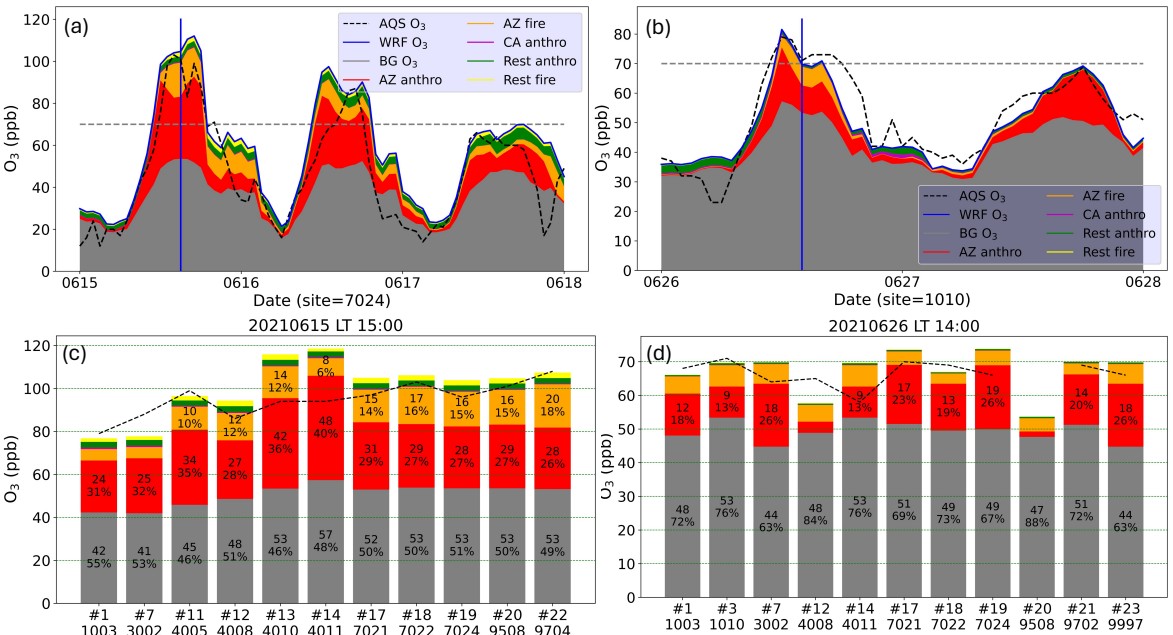

**Figure 12. Contribution of each tagged $O_3$ source to the hourly $O_3$ concentrations (ppb) for Case I (a-b) and Case II (c-d). The top panels (a, c) show the hourly variations in $O_3$ concentrations at a single AQS site (#7024 for Case I and #1010 for Case II) from 15-18 June 2021 and 26-28 June 2021, respectively. The bottom panels (b, d) display the contributions of different $O_3$ sources at multiple sites at the time stamps indicated by the blue vertical lines in (a) and (c). $O_3$ sources include background $O_3$ (BG $O_3$), Arizona anthropogenic (AZ anthro), California anthropogenic (CA anthro), rest of the anthropogenic (Rest anthro), Arizona fire (AZ fire), and rest of the fire (Rest fire).**

We also present in Figure 13 the WRF-Chem simulated surface CO concentrations on 15 June 2021 (Case I). By comparing the difference in CO concentrations with fire emissions (Figure 13b) and without fire emissions (Figure 13a) to the CO fire tag (Figure 13d), we observe a similar spatial pattern to that of $O_3$ in Figure 11. However, the CO fire tag indicates a more extensive area of low CO concentration coverage compared to the sensitivity method. The negative CO values observed in the sensitivity test (panel c) differ from the negative $O_3$ values, which are primarily driven by

nonlinear photochemical processes. Instead, negative CO values likely result from spatial and temporal variations in the CO plume caused by atmospheric transport and mixing. Specifically, shifts in plume location due to wind patterns and turbulent mixing can create regions where the modeled fire-related CO contributions are lower than the surrounding background levels, leading to apparent negative values.

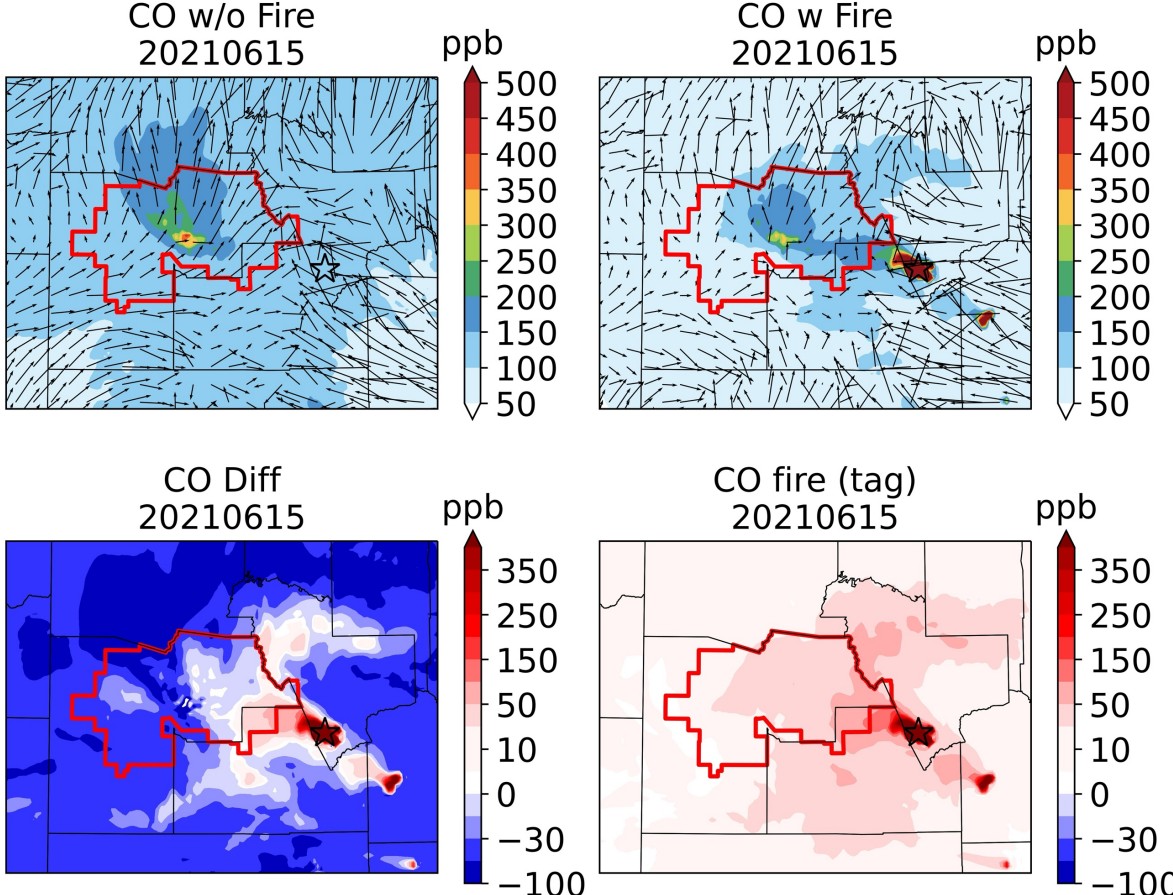

**Figure 13. Daytime (7:00-19:00 LT) average surface CO concentrations superimposed with wind vectors simulated by WRF-Chem during Case I (15 June 2021) for (a) without fire emissions, (b) with fire emissions, (c) difference between (b) and (a), and (d) CO fire tags. Stars mark the wildfire locations (Telegraph Fire at the top and Rafael Fire at the bottom). The red outline denotes the designated nonattainment area. Please refer to Figure S8 for wind speed contours of wind vectors at 1600 local time.**

**Impact of Fire on Chemistry and Meteorology.** We show in Figure 14 the temporal variations in the photolysis rate of $NO_2$ ($J_{NO2}$), $NO_X$ (NO+NO$_2$) $HO_x$ (OH+HO$_2$), and $O_3$ concentrations in metro Phoenix (Site 7024) at local time 16:00 over a seven-day period in June 2021, covering Case

I under two conditions: with and without fire emissions. This site (see Figure 1) is situated along the plume coverage downwind of the fire. We also included key meteorological variables (net and outgoing longwave radiation, winds, surface temperature and PBL height) and concentration of black carbon aerosols (which is a light absorbing particle) to elucidate the direct radiative impact of the fires. In Figure 14(a), the photolysis rates of $NO_2$ ($J_{NO2}$) are consistently only slightly higher without fire emissions while $NO_X$ concentrations vary across the week (lower in June 14 but slightly higher in June 15 with fire). $HO_X$ levels vary similarly with $NO_X$ during this fire event, possibly associated with VOCs from fires. This variation is consistent with $O_3$ plume chemistry (Robinson et al., 2021; Xu et al., 2021) where this variation results to $O_3$ in June 15 at 4pm that is significantly higher in the simulation with fire compared to simulation without fire. The net and outgoing longwave radiation, along with black carbon concentration are also higher with fire indicating more absorption of downward radiation similar to fire BC and OC impacts discussed in Jiang et al. (2012). Note that there is a significant wind shift from northward to southward (along with lower wind speed) in June 15 when fire is included (Figure 14d), resulting in the displacement of $O_3$ and CO hotspot observed in Figure 10 and Figure 13, respectively. This is consistent with the observed exceedance of $O_3$ levels on the same day. The simulated wind speed reduction at Site 19 (#7024) from 5.6 m/s to 2.1 m/s aligns with observed wind speeds at nearby sites, including Site 1 (#1003) at 2.1 m/s, Site 8 (#3003) at 2.0 m/s, and Site 11 (#4005) at 1.2 m/s. Similarly, the wind direction shifts from northward to southward is also captured in the simulations, as illustrated in Figures S9 and S10.

To further investigate the fire impact, we present in Figure 15 a cross-sectional view of the smoke plume as it travels towards Phoenix during Case I, highlighting the concentrations of multiple atmospheric pollutants, including $O_3$, CO, $NO_X$, HCHO (formaldehyde), $PM_{2.5}$ (particulate matter), and PAN (peroxyacetyl nitrate). Near the fire location, concentrations of CO, $NO_X$, HCHO, and $PM_{2.5}$, which are primary pollutants directly emitted from the fire, are high, whereas $O_3$ concentrations are lower. As the smoke moves closer to the urban region, $NO_X$ levels in the boundary layer increase significantly, along with $O_3$ levels reaching 100 ppb above the ground. Levels of $NO_X$ from fires diminishes at a faster rate than HCHO and $PM_{2.5}$ levels along the

trajectory. It is clear from the figure that pollutants from fires are transported towards the valley.

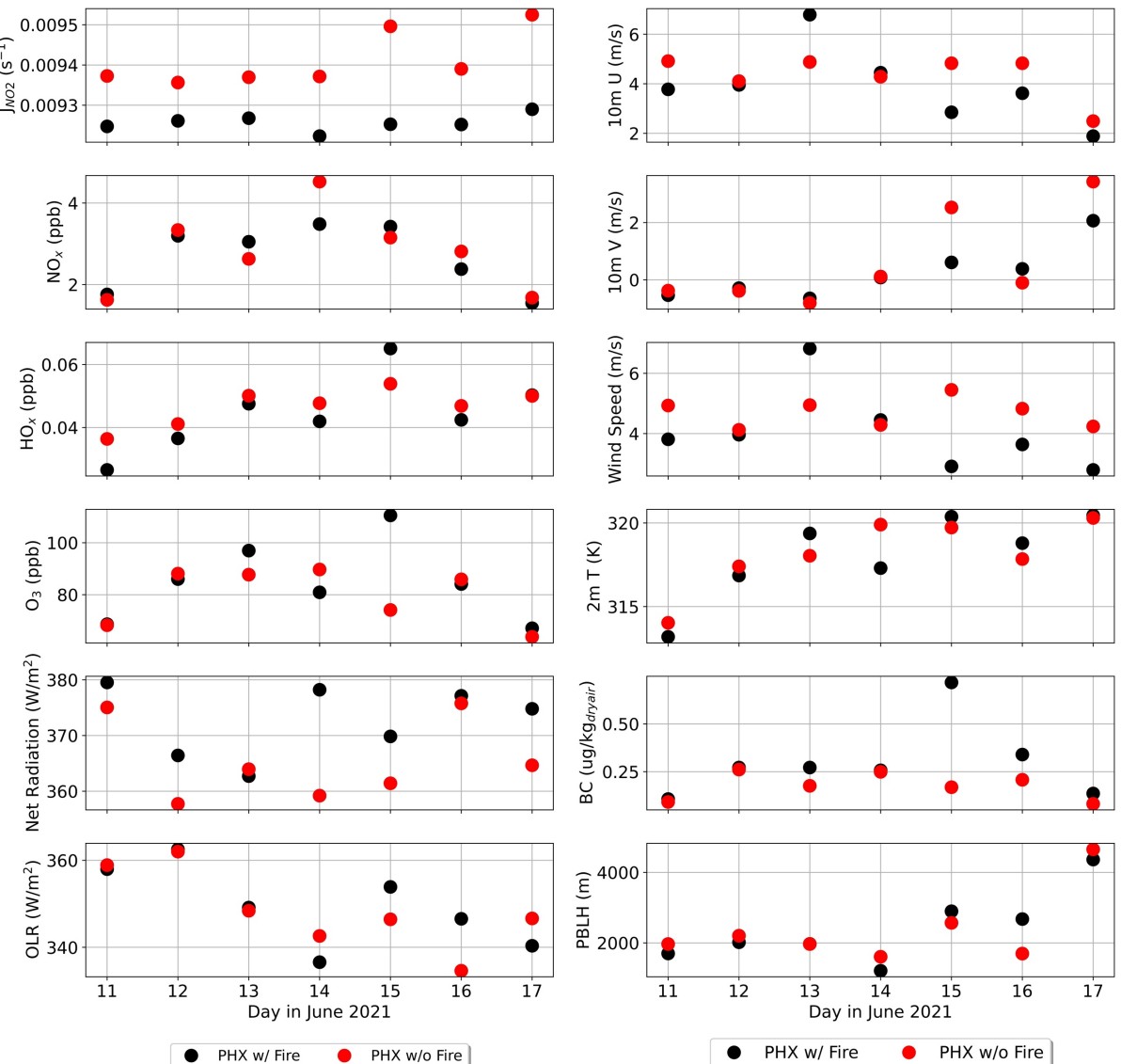

**Figure 14. WRF-Chem simulated time series (11-17 June 2021) of daily photolysis rate of NO₂ (J_NO2), concentrations of NO_X (NO+NO₂; ppb), HO_X (OH+HO₂; ppb), and O₃ (ppb), meteorological conditions such as net and outgoing longwave (OLR) radiation (watts/m²), 10-meter zonal and meridional and zonal wind and wind speed (10m U and V, Wind Speed (m/s), 2-meter air temperature (K), concentration of black carbon (BC) aerosols (μg/m³), and planetary boundary layer height (PBLH). All these are sampled at 16:00 local time in Phoenix (Site 19 - #7024). The black markers represent the values with fire emissions, while the red markers indicate values without fire emissions.**

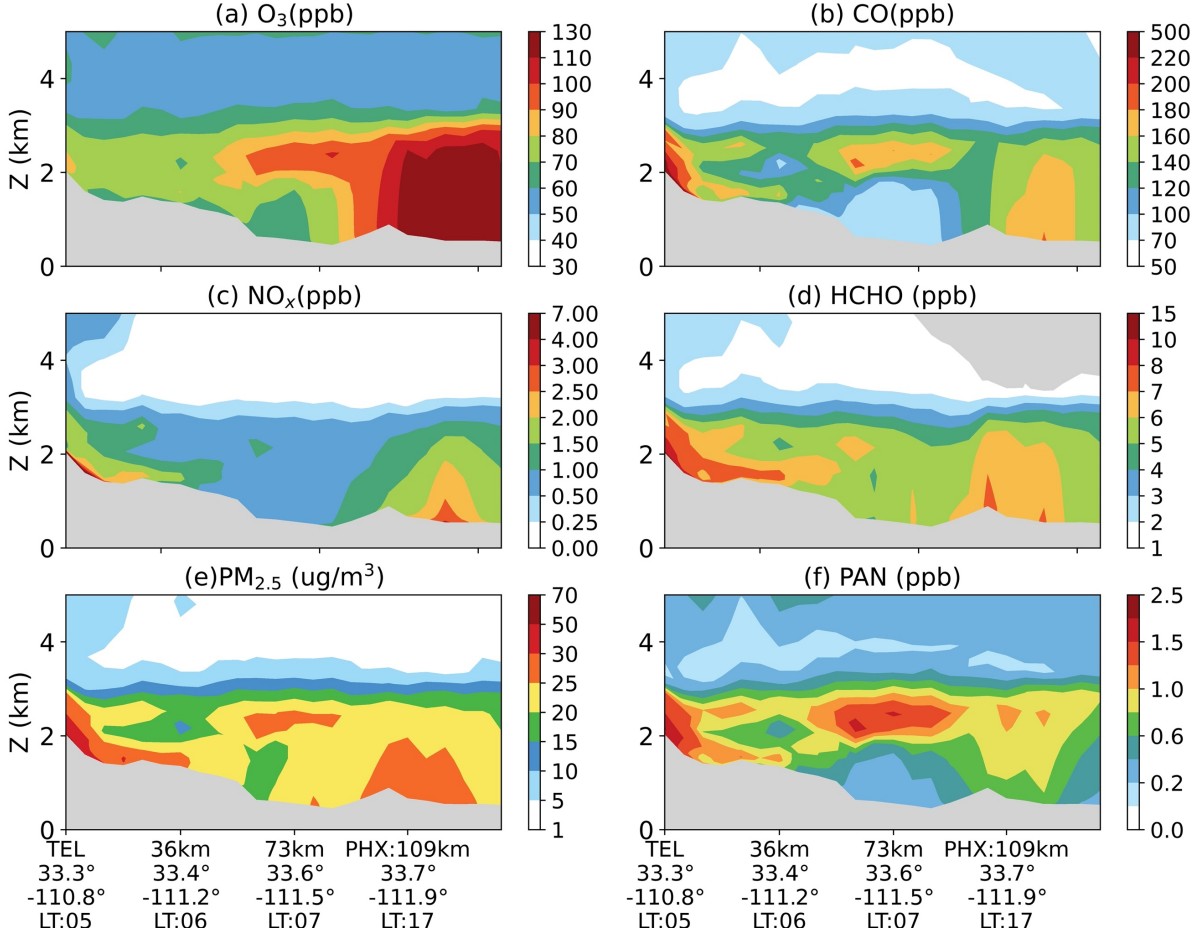

**Figure 15. Cross-sectional analysis of a smoke plume traveling towards Phoenix during 15 June 2021, showing vertical and horizontal distribution of various pollutants. The subplots represent concentrations of (a) O₃, (b) CO, (c) NOₓ, (d) HCHO, (e) PM₂.₅, and (f) PAN (Peroxyacetyl Nitrate) across different altitudes and distances from the fire where TEL means Telegraph. The plume path is denoted in Figure 11. The gray shading represents the topography heights.**

This is particularly true for PAN which shows an enhancement above the valley along with CO and PM$_{2.5}$. These enhancements aloft are not present in the cross-section of WRF-Chem simulation without fire emissions (see Figure S7). Previous studies have indicated that the rapid conversion of NO$_X$ to PAN can limit O₃ production near fires, especially at low temperatures, but the decomposition of PAN can lead to additional O₃ production further downwind of the fires especially in the presence of higher amounts of VOCs (Alvarado et al., 2010; Jaffe et al., 2013). The concentrations of O₃, CO, NO$_X$, and PAN from fire tags presented in Figure S6 (alongside

Figure S7) further demonstrate that fire smoke exacerbates urban $O_3$ levels, while the exceedance is predominantly from local production.

In summary, this cross-sectional analysis illustrates the complex vertical and horizontal distribution of various pollutants and their transformations within a smoke plume traveling towards Phoenix. The interaction between primary emissions from fires, secondary pollutants formed during transport, and the presence of local anthropogenic emissions in the urban environment highlights the multifaceted nature of urban air quality impacts during wildfire events.

**Fire-induced Changes in Chemical Regime.** We show in Figures 16 and 17 the associated impact of fires on the chemical regimes of $O_3$ formation over Phoenix at local time 14:00. Here, two key observable indicators are chosen to illustrate this impact: the HCHO/NO₂ ratio, also known as the Formaldehyde to $NO_2$ Ratio (FNR), and the $O_3/NO_X$ ratio (Sillman, 1995; Tonnesen et al., 2000; Zhang et al., 2009). The HCHO/NO₂ ratio (FNR) has been used in previous studies as an indicator for determining the sensitivity of $O_3$ formation to either VOCs or $NO_x$ (Martin et al., 2004; Jin et al., 2020; Mirrezaei et al., 2024). Zhang et al. (2009) recommended a transition value for surface FNR of 1 in agreement with Tonnesen and Dennis (2000) and Martin et al. (2004). For tropospheric column FNRs, Jin et al. (2020) recommended a transition range of 3.2–4.1, which has been successfully validated in recent studies, showing good agreement with chemical regimes derived from surface measurements. A higher FNR than transition indicates a $NO_x$-limited regime, where $O_3$ formation is more sensitive to changes in $NO_x$ emissions, while a lower FNR than transition points to a VOC-limited regime, where $O_3$ formation is more responsive to changes in VOCs. In the context of wildfire smoke, the influx of VOCs from the fires can shift the chemical regime from VOC-limited to $NO_X$-limited, altering the dynamics of $O_3$ production in the urban area (Jin et al., 2023; Robinson et al., 2021; Rickly et al., 2023). This shift can lead to unexpected increases in $O_3$ levels as the balance of precursors is altered by the incoming smoke plume. To complement the primary HCHO/NO₂-based indicator of ozone production, we also explored $O_3/NO_X$ ratio as an additional indicator to provide context regarding the oxidizing capacity of the atmosphere and the contributions of various chemical pathways to $O_3$ production. The recommended transition value for $O_3/NO_X$ ratio (60) by Zhang et al. (2009) is significantly larger than originally suggested by Tonnesen and Dennis, (2000). A higher ratio suggests an environment with abundant VOC oxidation, often associated with high levels of $O_3$ production. Conversely, a lower ratio indicates a dominance of $NO_X$ oxidation pathways, which can suppress $O_3$ formation under certain

conditions. Due to variations in transition values, we will focus solely on changes in this $O_3/NO_X$ ratio during a fire event compared to a no-fire event.

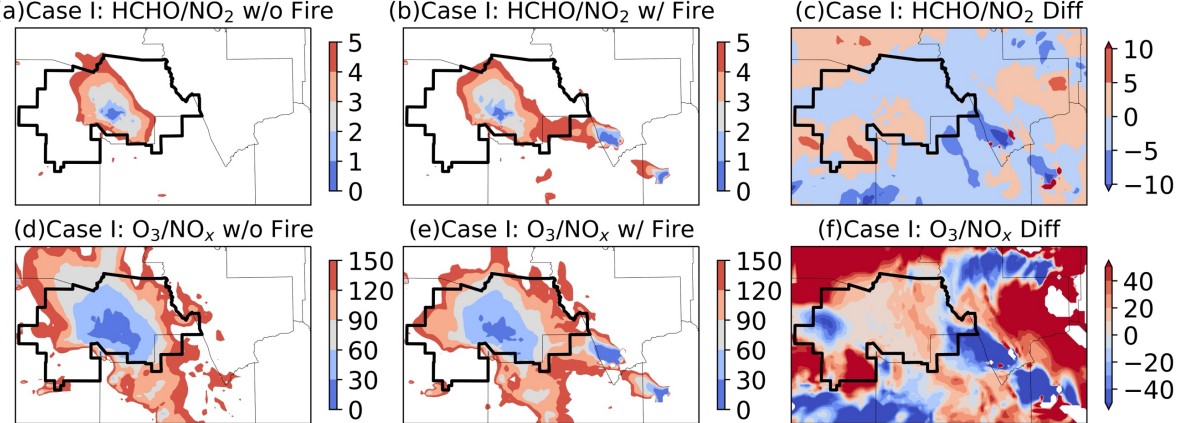

**Figure 16. Sensitivity analysis of different ratios in Case I at local time 14:00 under conditions without and with fire. The top row (a-c) is the surface HCHO/NO₂ ratio, while the bottom row (d-f) depicts the surface $O_3/NO_X$ ratio. Ratios larger than 5 for HCHO/NO₂ (150 for $O_3/NO_X$ ) are shown as white space. (a) and (d) show the respective ratios without fire, and (b) and (e) display the ratios with fire. (c) and (f) represent the differences in these ratios between the scenarios with and without fire. The red color in (c) and (f) indicates a shift towards a more $NO_x$-limited regime, and blue indicates a shift towards a more VOC-limited regime.**

The presence of wildfire emissions can increase the levels of both VOCs and $NO_x$, thereby influencing these ratios and providing insights into the changing oxidative environment over Phoenix. The relative change in VOCs and $NO_x$ will affect $O_3$ sensitivity, depending on which of these pollutants has a larger percentage change relative to its current levels. Miech et al. (2024) found that at Phoenix JLG supersite, when the sensitivity is under VOC-limited, FNR is higher

than normal suggesting elevated VOCs relative to $NO_2$ under a smoke event and shifting the sensitivity towards a transitional or $NO_X$-limited. This is also seen in Figure 15 where levels of CO and HCHO are relatively elevated than $NO_X$ along the fire plume trajectory.

In Figures 16 and 17, the analysis of these two surface ratios reveals how wildfire smoke alters the chemical regime over Phoenix at local time 14:00 when $O_3$ production is expected to peak. Without

the smoke plume, the majority of Phoenix urban area in the early afternoon, when the photolysis is highest, is already under a transitional/$NO_x$-limited regime (Figure 16a). With the presence of smoke, additional $NO_x$ and VOCs are brought to the region and the regime shifts towards more

NOx-limited in the central urban region, as seen by the increase (orange contours) in the FNR (Figure 16c), consistent with Miech et al. (2024). In contrast, FNR decreases across the broader extent of the fire, (blue contours), most likely with the introduction of $NO_X$ from PAN decomposition further downwind. Comparisons of tropospheric column FNRs (Figure 17) show that WRF-Chem effectively captures the wildfire event occurring at the Phoenix urban interface, agreeing well with TROPOMI observations and accurately representing the plume trajectory extending toward Phoenix Metro. Notably, the surface FNR pattern closely mirrors the column FNR, indicating that WRF-Chem successfully captures shifts in the chemical regime during fire events.

The impact of the wildfire varies across different areas of central Arizona. In central Phoenix, where $NO_X$ levels are already high, the fire's influence on FNR is less pronounced, despite increased HCHO levels. In contrast, in suburban areas along the plume pathway, such as Gilbert, Mesa, and Chandler—where conditions are more $NO_X$-limited to transitional—FNRs tend to decrease, shifting the chemical regime toward a less $NO_X$-limited state. These spatial variations are critical, as wildfire events near large urban centers interact with existing local emissions, significantly influencing $O_3$ formation dynamics. This understanding is especially valuable for compound events, such as the wildfire-heatwave scenario examined in this study, where both factors contribute to $O_3$ exceedances. The $O_3/NO_X$ ratio shown in Figure 16d–f further supports this, revealing slightly increasing ratios in the metropolitan area and significantly decreasing ratios near the fire source. Additionally, comparisons with surface $O_3/NO_X$ ratios at the JLG supersite (though limited) indicate that the simulated shift toward slightly higher ratios agrees better with observed $O_3/NO_X$ trends, reinforcing the model's effectiveness in capturing fire-induced chemical variations consistent with fire and smoke modeling and observational studies (Buysse et al. 2019; Rickly et al., 2023; Robinson et al., 2021; Xu et al., 2021; Jin et al., 2023; Holder and Sullivan, 2024; Guo et al., 2017 among others).

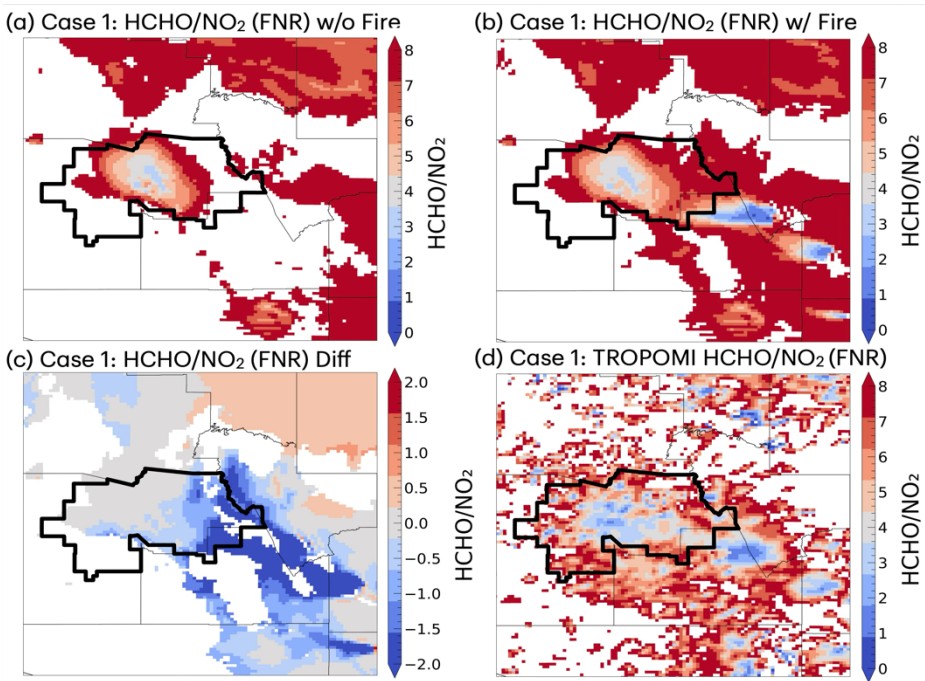

**Figure 17. Comparison of columnar HCHO/NO₂ ratio (FNR) between WRF-Chem without (a) and with fire (b) and TROPOMI (d) for Case 1. Ratios larger than 8 are shown as white space. Panel (c) correspond to the difference of WRF-Chem FNRs between with fire and without fire. The red color in (c) indicates a shift towards a more NOₓ-limited regime (higher FNR with fire), and blue indicates a shift towards a more VOC-limited regime (lower FNR with fire). Note that the range of FNR for a transitional regime is 3.2 to 4.1 based on Jin et al. (2020).**

## 4. Summary and Future Directions

This study investigates the impact of wildfire smoke on surface $O_3$ chemistry and urban meteorology in the Phoenix metropolitan area during June 2021, a month marked by compounding extreme events. This period saw an unprecedented heatwave and widespread wildfires at the urban interface, driven by severe drought conditions and a persistent high-pressure system. These overlapping extremes created a unique and challenging environment for air quality and

atmospheric dynamics, making Phoenix, Arizona a critical case study for understanding wildfire induced $O_3$ pollution in urban settings. We apply model sensitivity and species tagging approaches in the high-resolution configuration of the WRF-Chem model to quantify its contribution relative to anthropogenic local emissions and regional transport. We especially highlight the utility of combining these modeling frameworks, along with comparisons to surface and satellite

observations, in elucidating their impacts given the advantages of a finer scale coupled (weather-air quality) representation of these compound events in complex terrain and arid environments. Two specific cases (Telegraph and Rafael fires) are chosen for analyzing the wildfire impact in Phoenix, Arizona. During these cases, the MDA8 $O_3$ is observed to exceed the NAAQS standard (70 ppb), especially for case I (Telegraph fire), with concentrations reaching 110 ppb.

Overall, the $O_3$ levels in Arizona are influenced by a combination of background levels, local anthropogenic emissions, and wildfire contributions. The highest $O_3$ levels were observed around urban centers, with wildfires significantly contributing to elevated $O_3$, especially near the fire sites and downwind areas. The spatial and temporal distributions of CO and $O_3$, as well as the contributions from different tags in Arizona during the wildfire season, reveal significant

contributions of both anthropogenic and wildfire emissions to CO levels across the state, with local urban emissions still playing a dominant role in areas like Phoenix and Tucson. However, wildfire emissions were particularly impactful in regions downwind of the fires.

In fact, our simulations show that wildfire emissions notably increased the MDA8 $O_3$ levels during two fire plume case studies that we examined. The Telegraph Fire, in particular, contributed to

significant $O_3$ levels on June 15. The results demonstrated that background $O_3$ levels account the bulk of total $O_3$ (around 50%), with local anthropogenic emissions contributing significantly (24% to 40%) depending on the urban setting. Our background $O_3$ estimate is consistent with recent reports (e.g., Parrish et al. 2025). During peak $O_3$ hours, fire-contributed $O_3$ was significant across multiple sites, ranging from 5 to 23 ppb or 5% to 21 % of total $O_3$ levels, with an average of 15

ppb or 15%. Without smoke, the Phoenix urban area is primarily under a transition to $NO_x$-limited regime in the early afternoon when $O_3$ photolysis rate is highest. With smoke present, the central urban region becomes more $NO_x$-limited due to the addition of VOCs transported from the fires relative to $NO_x$ which is already high from local anthropogenic emissions. In contrast, the suburban and rural areas downwind of the fires generally experience a decrease in the ratio shifting towards

a more VOC-limited regime, which is likely due to the addition of $NO_X$ from fires as a result of thermal decomposition of PAN from the fires transported to these areas. Furthermore, our findings indicate that smoke influenced local wind speed and direction in Phoenix, leading to subtle shifts in the spatial distribution of pollution levels.

By closely investigating these tags, we also find differences between Phoenix and Yuma. Unlike Yuma, where $O_3$ levels are significantly influenced by transboundary emissions from California and Mexico, Phoenix's $O_3$ levels are primarily driven by local emissions, with much smaller contributions from these external sources during the study period. Specifically for a smoky day, during the diurnal cycle of $O_3$ levels, anthropogenic emission contributed local $O_3$ production dominate in the early morning and early afternoon, while fire related $O_3$ contributions increase significantly in the late afternoon when a smoke plume passes through. This pattern is observed across suburban, urban, and rural settings, with fire related $O_3$ sometimes surpassing local anthropogenic production in rural areas.

Although our findings are reasonably consistent with previous studies and available (albeit limited) observations, it is important to acknowledge several limitations in our analysis. First, we only tagged $NO_x$ emissions, which may not fully capture $O_3$ dynamics in VOC-limited regions where VOCs play a more significant role in $O_3$ formation. Additionally, the limited number of cases analyzed reduces the generalizability of the results. Expanding the study to include more cases would improve robustness. Furthermore, the absence of vertical profile data limits the depth of the analysis, particularly for understanding pollutant distributions in the upper atmosphere. In the future, with more spatial coverage, including diurnal variation data from Tropospheric Emissions: Monitoring of Pollution (TEMPO), would enhance the temporal and spatial representation of $O_3$ and its precursors. Refinements in fire emission estimates from FINN, plume rise calculations, and the validation of aerosol direct radiative effects on dynamics and thermodynamics, particularly radiation fluxes would increase model accuracy. Lastly, incorporating more detailed sectoral tags would better differentiate emission sources and improve $O_3$ and aerosol impact assessments. Moreover, since tagged $O_3$ is represented as tracers in the model, its production and immediate loss are primarily captured, while some loss processes may not be fully accounted for. Processes such as detailed fire-plume shading effects and nighttime $O_3$ titration chemistry may therefore be underrepresented.

As has been suggested by previous studies, the substantial enhancements in $O_3$ concentrations due to wildfire emissions highlight the necessity of accounting for wildfire impacts in formulating effective air quality management strategies. Such strategies should consider the influence of fire emissions on urban $O_3$ and more notably their subsequent interactions with local emissions, chemistry, and meteorology (e.g., He et al., 2024) to help provide additional perspectives on

current $O_3$ pollution assessments. This is especially the case over urban areas in semi-arid/arid environments like southwest United States, where confounding (and compounding) factors arising from meteorological extremes and dynamical challenges due to complex topography are present; notwithstanding the effect of climate change to increasing global aridity (European Commission: Joint Research Centre et al., 2018), fire frequency and intensity (e.g., Jones et al., 2022), biogenic activity (Jiang et al. 2018, Pfannerstill et al. 2024), and anthropogenic VOCs (Qin et al., 2024). Integrating source attribution approaches, including data-driven techniques, within coupled weather-air quality models, along with a more robust observational infrastructure, can serve as a valuable complement to existing methods for improving our understanding of $O_3$ dynamics. Strengthening these efforts will enhance the accuracy of pollution source identification and support more effective air quality management strategies.

*Code and Data Availability Statement.*

The WRF-Chem model is publicly available at https://github.com/wrf-model/WRF (last access: 25 June 2022). The model outputs can be provided upon request to the corresponding author. EPA AQS and PAMS hourly and daily datasets are available at https://aqs.epa.gov/aqsweb/airdata/download_files.html.

*Author contributions.*

YG and AA designed the research. YG performed the model runs and subsequent analysis. YG wrote the paper with contributions from AA and AS. AM helped with the satellite data analysis.

*Competing Interests.*

Some authors are members of the editorial board of journal Atmospheric Chemistry and Physics.

*Acknowledgments.*

We especially acknowledge Dr. Gabriele Pfister and Dr. Louisa Emmons at NCAR/ACOM for kindly providing the guidance for implementing tagging in WRF-Chem. We also greatly appreciate the code availability shared with the public through Lupaşcu et al. (2019), which this study was built upon. We also thank Dr. Matthew Pace, Dr. Rene Nsanzineza, and Michael Graves at the Arizona Department of Environmental Quality (ADEQ) for the help with observations for the state of Arizona.

*Financial Support.*

This work is supported by the Arizona Board of Regents (ABOR) Grant from the Technology and

Research Initiative Fund (TRIF).

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
