# Peer review of "Source contribution to ozone pollution during June 2021 fire events in Arizona: Insights from WRF-Chem tagged O3 and CO"

_EGUsphere, 2024_

## Author Comment (AC2)

Authors' comments in response to all comments made in the open discussion phase.

We thank the two Anonymous Referees for their thoughtful and helpful comments on our submission. We have revised our manuscript taking all these comments into account. Here we repeat the comments in *italic font*, and in each case provide our responses and a summary of the resulting changes to the manuscript (if any) in normal blue font.

Reviewer #1

*Review of "Source contribution to ozone pollution during June 2021 in Arizona: Insights from WRF-Chem tagged O3 and CO" by Guo et al.*

*This is a study on the source attribution of surface and tropospheric O3 and CO in Arizona during one month (June 2021) when fire emissions impacted the area. The authors use WRF-Chem tagged simulations as well as WRF-Chem sensitivity simulations (with and without fire) to determine the budget of O3 for the region, including the fire contribution. They also investigate the meteorological, diurnal, and chemical conditions that influence O3 production for two case studies in detail. While the study covers only a small spatial and temporal region/period, this kind of detailed case study is very interesting as there remains a high level of uncertainty and variability in O3 production during fire events. My comments below are mainly related to some missing information and discussion that, if added, would make the results clearer and put into better context.*

*Line by line comments:*

*Lines 60-64: The wording could be improved since it's not the vehicles or industrial buildings that are combusting. Better to say fossil fuel combustion by vehicles, industry, and power plants. Also add "natural" in "as well as the natural biogenic emissions" since the start of the sentence says anthropogenic activities.*

Thank you for the feedback. We have revised the wording to be clearer accordingly.

*Lines 75-86: I realize this isn't supposed to be an exhaustive list, but you could add GEOS-Chem, which has had O3 tagging options for quite some time. Described originally in Wang et al (1998) and updates in Zhang et al (2008) and used in several studies, including Whaley et al (2015), for example.*

*Wang, Y., and D. J. Jacob (1998), Anthropogenic forcing on tropospheric ozone and OH since preindustrial times, J. Geophys. Res., 103, 31,123–31,135, doi:10.1029/1998JD100004.*

*Zhang, L., et al. (2008), Transpacific transport of ozone pollution and the effect of recent Asian emission increases on air quality in North America: An integrated analysis using satellite, aircraft, ozonesonde, and surface observations, Atmos. Chem. Phys., 8, 6117–6136, doi:10.5194/acp-8-6117-2008.*

*Whaley, C. H., K. Strong, D. B. A. Jones, T. W. Walker, Z. Jiang, D. K. Henze, M. A. Cooke, C. A. McLinden, R. L. Mittermeier, M. Pommier, et al. (2015), Toronto area ozone: Long-term*

*measurements and modeled sources of poor air quality events, J. Geophys. Res. Atmos., 120, 11,368–11,390, doi:10.1002/2014JD022984.*

Thank you for the suggestion. We have incorporated GEOS-Chem into the list with the corresponding references Wang et al. (1998), Zhang et al. (2008), Whaley et al. (2015) as:

"…, the global chemical transport model (CTM) with assimilated meteorological observations from the Goddard Earth Observing System (GEOS-Chem) (Wang et al. (1998), Zhang et al. (2008), Whaley et al. (2015)), …"

*Section 2: can you include information about what kinds of vegetation are in this region that burned in the wildfires?*

Thank you for the suggestion. We have added the vegetation type for both wildfires as:

"The burn area was located in the southernmost region of Tonto National Forest, primarily characterized by desert shrubs and grassland vegetation (USDA, 2024)."

"The Rafael Fire had burned over 38 square miles by late June in the Coconino National Forest, where evergreen shrubs were the dominate vegetation type (Conservation Biology Institute, 2024)."

References:

*Conservation Biology Institute, 2024: Potential Natural Vegetation Type Dataset, Data Basin, https://databasin.org/maps/new/#datasets=43a107f2f0c048f8a87a97adf0368ee9 (last access: 9 November 2024).*

*USDA Forest Service, 2024: Tonto National Forest - Nature & Science, USDA Forest Service, https://www.fs.usda.gov/detail/tonto/learning/nature-science/?cid=fsbdev3_018777 (last access: 9 November 2024).*

*Figure 1a: Can an outline of the city of Phoenix be marked on the left panel? The caption implies that this whole area is considered Phoenix.*

Thank you for the suggestion. We have added an outline of the Phoenix-Mesa-Scottsdale Metropolitan Statistical (MSA) area on the left panel of Figure 1a to clearly indicate its boundaries. We have also revised the caption to specify that the outlined area represents Phoenix while the larger area shows the surrounding region. The clarification is also added in section 2.

*Section 2.2: Does the WRF-Chem model include the radiative effects of smoke on the air temperature and the photochemical production of O3? E.g., when smoke is heavy and shading the surface, does O3 production and surface temperature go down? If so, please say explicitly, as not all models do this. And if not, then this is a caveat to all of your O3 results that come in Section 3.2, as it's an important missing process.*

Thank you for your insightful question. Yes, the WRF-Chem model includes the radiative effects of aerosols, such as smoke, on air temperature and photochemistry, capturing both direct and indirect impacts on atmospheric processes. Our simulations were performed with both the

direct and indirect effects turned on. Specifically, the model accounts for aerosol radiative feedback, meaning that when smoke is present, its optical properties, such as absorption and scattering, are included in radiative transfer calculations. Heavy smoke can significantly reduce the solar radiation reaching the surface, leading to a cooling effect on both the surface and the boundary layer. Additionally, smoke can decrease the intensity of UV radiation, which is crucial for photochemical $O_3$ production.

We have added this information in section 2.2.

*Figure 2: how come you don't include TROPOMI O3 and CO as well?*

Thank you for your suggestion. We have included TROPOMI CO in Figure 2 for reference. We have not used TROPOMI $O_3$ data because the high-resolution TROPOMI $O_3$ product is total column $O_3$ which mostly impacted by stratospheric $O_3$ rather than tropospheric levels. Additionally, the tropospheric column $O_3$ product is only available for latitudes between -20° and 20° which does not cover our study domain, as its retrieval is based on the Convective Cloud Differential (CCD) method. This method is more effective in the tropics due to the frequent occurrence of high convective clouds. Although $O_3$ profile product is available, it requires algorithm for additional processing to calculate the tropospheric column and further validation, which might be outside the scope of this paper. Also, the $O_3$ profiles do not represent the surface $O_3$ unlike $NO_2$ who has shorter lifetime and dominant source from combustion.

We have listed some related references here and revised section 2.6 accordingly.

1. Copernicus Sentinel data processed by ESA, German Aerospace Center (DLR), Sentinel-5P TROPOMI Total Ozone Column 1-Orbit L2 5.5km x 3.5km, Greenbelt, MD, USA, Goddard Earth Sciences Data and Information Services Center (GES DISC), Accessed: 17 November 2024., 10.5270/S5P-ft13p57, 2020.
2. Copernicus Sentinel data processed by ESA, German Aerospace Center-Institute for Environmental Research/University of Bremen (DLR_IUP), Sentinel-5P TROPOMI Tropospheric Ozone Column, Greenbelt, MD, USA, Goddard Earth Sciences Data and Information Services Center (GES DISC), Accessed: 17 November 2024., 10.5270/S5P-hcp1l2m, 2020.
3. Heue, K.-P., Eichmann, K.-U. & Valks, P., *TROPOMI/S5P ATBD of Tropospheric Ozone Data Products.* https://sentinels.copernicus.eu/documents/247904/2476257/Sentinel-5P-ATBD-TROPOMI-Tropospheric-Ozone.pdf/d2106102-b5c3-4d28-b752-026e3448aab2?t=1625507455328 Accessed 17 November 2024, 2021.
4. Cazorla, M., & Herrera, E., An ozonesonde evaluation of spaceborne observations in the Andean tropics. *Scientific Reports*, *12*(1), 15942, 2022.

*Figure 4: Similarly, why is TROPOMI O3 not shown in Figure 4?*

Please refer to the response above.

*Fig 4 vs Fig S2: These two figures should either (a) have the same colour scales per species, so that we can more easily see the difference between the two case studies, or (b) have colour scales that highlight the spatial distribution of the enhancements. The HCHO row does both (a&b), the CO row does the latter (just b), but the NO2 row seems to do neither. Can you please adjust the colour bars of the NO2 row in these figures to at least do one or the other?*

Thank you for the suggestion. We have adjusted the colour scales of $NO_2$ and CO to be consistent across both cases for easier comparison. Initially, we set different color bars for Case I and Case II due to the significantly higher concentrations in Case I. However, we agree that a consistent colour scale is beneficial for direct comparison between cases, and we have made this change accordingly.

*Fig 5 & Fig S3: Ok, now I see that the colour scales above were set to match those for these model figures which follow. If that's the priority, then I suggest that you re-jig these figures so that the TROPOMI and WRFChem tropospheric columns appear right on top of each other (e.g. one row for TROPOMI HCHO, and next row for modelled HCHO). You could potentially also add a model-minus-satellite row as well to better see the differences..*

Thank you for this valuable suggestion. We have revised the color bars in Figure 5 to maintain consistency between TROPOMI and WRF-Chem simulations across both cases.

Regarding the direct overlay of TROPOMI and model outputs, we chose not to align them row-by-row due to the differing spatial and vertical resolutions and grid structures (TROPOMI at about 7 km and WRF-Chem at 3 km), which could compromise the modelled high-resolution data. Additionally, displaying HCHO, $NO_2$, and CO together in a single figure enables us to highlight the smoke/fire signatures more effectively.

*Line 351: I believe you mean Figures S3-S4 here.*

Thank you for catching that. It should indeed refer to Figures S3–S4. We have made this correction in the text.

*Lines 390-395: This background information seems out of place. You could have included this when you first mention CO in the paper. Some of it (e.g., that CO is emitted when there is incomplete combustion) you can presume the reader already knows.*

Thank you for your comment. We have revised this paragraph and moved it to Section 1 in the paper as first introduction of CO.

*Line 402-404: Is the background O3 really an "absence of local sources"? Or are local natural fluxes (e.g., isoprene, soil NOx, lightning NOx, stratospheric O3 decent) included to contribute to this high background O3 during the heatwave? I think you probably mean absence of local anthropogenic sources... and maybe you can clarify whether biogenic and natural sources are included in panel (b). Line 404 says that the background O3 is due only to "regional and global influences on a monthly basis", which the reader could interpret as only long-range anthropogenic sources.*

Thank you for this insightful comment. We apologize for the ambiguity and are happy to clarify what we mean by "background O₃" in our analysis.

By "background O₃", we refer to the residual O₃ after accounting for contributions from anthropogenic and fire emissions, rather than implying a complete "absence of local sources". In our analysis, the background O₃ in panel (b) includes both natural contributions and long-range transport, which encompass both natural and anthropogenic components. We have clarified in section 3.2 that "background" refers to total O₃ minus tagged contributions from anthropogenic and fire emissions, explicitly including both local and transported contributions from natural sources.

*Line 410-411: A few additional words can help clarify this sentence: "Mexico's anthropogenic contributions \*to O3\* (Figure 8f) have a larger impact than \*they do for\* CO (Figure 7f)…"*

Thank you for the suggestion. We have revised this sentence by specifying "Mexico's anthropogenic contributions to O₃ (Figure 8f) have a larger impact than they do for CO (Figure 7f) in terms of spatial coverage, affecting most of the southern Arizona regions and even reaching Phoenix at 3 ppb. The magnitude is also higher for O₃, reaching 10 ppb in Yuma.", as recommended.

*Figure 8: I notice here and in other O3 figures, there is never a negative contribution to O3 concentrations, which could occur if, for example, the emission source caused O3 titration or a transition on the O3 formation chemical regime, or if the fire emissions caused shading that reduced photochemical production of O3. Do those circumstances really never happen in the model in your study? Or have they just been averaged out in the June mean? Perhaps the authors could include in the discussion the fact that there is no reduction in modelled O3 from any source at any part of the time series and why that may be (missing process in the model?).*

Thank you for your insightful question. In Figure 8, we did not observe any negative contributions to O₃ concentrations for the month of June as Figure 8 represents the 90th percentile values rather than instantaneous O₃ concentrations. However, we have noted lower O₃ concentrations in this region, particularly during nighttime, when O₃ loss is more pronounced (please see Guo et al., 2024). It is important to clarify that the O₃ tags in our model are designed as tracers that represent O₃ production, including its chemical loss immediately after the production. These tagged O₃ values decay as they are transported downwind but do not reach negative values due to a non-zero constraint in the model during integration.

We acknowledge that this is indeed an important consideration, and we have added a discussion in the conclusion section to address this point. While shading from fire emissions could lead to reductions in photochemical O₃ production, we do find lower photolysis rates in the fire plumes in our model as shown in Figure 14(a). However, significant O₃ losses near emission sources are not apparent, likely because we are focusing on O₃ production during peak hours during the daytime. O₃ losses are expected to be more prominent at night, which are not evident in the June-averaged plots presented in our current figures.

*Line 431: "Here" should instead be "In Figure 9".*

Revised accordingly.

*Line 439-440: I'm not sure I see this in Figure 9. June 17th only seems remarkable in that it's the day when there is the least good match between the model and measured O3 in the time series, and so I don't think it should be emphasized. Also, the background O3 contribution, which may or may not include local natural sources (see comment above) is the dominant O3 source throughout this time series. The red, which represents the Arizona anthro contribution is also significant throughout, and in particular on June 15th, so I'm not sure why Mexico is emphasized in the text.*

Thank you for the comment. Our intent is to underscore that while the contributions from Mexican emissions may not dominate the total ozone concentration, they frequently play a significant role in pushing $O_3$ levels above the NAAQS 70 ppb threshold. This highlights the importance of transboundary pollution in influencing air quality in Yuma. For instance, on June 15, contributions from Mexico surpass those from Arizona, illustrating the substantial influence of cross-border emissions on ozone exceedances.

We have revised the text to better clarify the role of these contributions in the context of Yuma's $O_3$ exceedances.

*Lines 460-469: As per the comment above about what fire processes the model is including: Does it include the shading effects of the smoke on the O3 production? If not, then please include that in the discussion here as a caveat. If it were, the O3 increase with fires may not be so high (and may even decrease O3).*

Thank you for the insightful comment. The WRF-Chem model used in our study does indeed include the effects of smoke aerosols on photolysis rates, see previous response and section 2.2 in revised manuscript. The effects of smoke plumes on photolysis rates are also illustrated in Figure 14, which shows that the smoke can reduce sunlight reaching the lower atmosphere and decrease photochemical $O_3$ production. Therefore, the observed $O_3$ increase with fire events in our results already reflects these smoke-aerosol interactions.

*Figure 11: Can you please make the colour scales the same between panels (a) and (b). Similarly, make the colour bars the same for panels (c) and (d), as the purpose of the figure is to compare the O3 diff technique with the O3 tag technique.*

Thank you for this helpful suggestion. We have updated Figure 11 to use uniform color scales for both cases and scenarios to enhance visual consistency and facilitate easier comparison of the results. We have also included negative color scales to show the non-linearity of sensitivity method compared to the tagging technique.

*Lines 532-535: could you please add here what causes the negative values in Figure 13c? While I was expecting/looking for negative values to appear for O3, I don't know why negative values would appear for the fire influence on CO.*

Thank you for your observation. The negative values for the fire influence on CO in Figure 13c are indeed unlikely to result from photochemical processes. These negative values may arise from spatial and temporal variations in the CO plume, driven by atmospheric transport and

mixing processes. Specifically, shifts in the plume location due to wind patterns and turbulent mixing could lead to areas where the modeled fire contribution to CO is lower than the background levels, creating apparent negative values. Figure 14 provides evidence of such dynamics, showing a shift in wind direction and differences in wind speed, particularly on June 15. This highlights the complex interplay between plume dispersion and atmospheric processes in shaping the spatial distribution of CO.

*Lines 591-593: Don't you mean "...the influx of VOCs from the fires can shift the chemical regime from VOC-limited to $NO_x$-limited, altering…"?*

Thank you for pointing this out. Yes, over Phoenix, the difference shown in pink indicates a shift toward a $NO_x$-limited regime due to the influx of VOCs from the fires. We have revised the text to clarify this.

*Figure 16c and f: could you add in the caption which chemical regime red represents a move towards and which chemical regime blue is a move toward?*

Thank you for the suggestion. We have added in the caption of Figure 16.

Reviewer #2

*Guo et al. used WRF-Chem model to identify the source of ozone in summer of Arizona. Overall, the method is robust and the results are reliable. However, the novelty of this study is not fully revealed. Some conclusions seem to be well known. I suggest the authors should stress the major findings and the novelty of this study. The detailed comments are as follows:*

1. *The authors only introduced the importance of source attribution techniques, while this study lacks of the introduction of the novelty in Arizona compared with previous studies in California and many other regions.*

   Thank you for your insightful comment. The novelty of our study lies not only in the application of source attribution techniques to Arizona but also in the unique environmental and atmospheric conditions of the region. Arizona's dry summer and monsoon summer create distinct shifts in the $O_3$ chemical regime (see our previous study Greenslade et al., 2024), as the monsoon season introduces additional moisture, influencing both the dynamics of wildfire smoke and the overall atmospheric chemistry. Moreover, Arizona's urban pollution island, particularly over Phoenix, differs from those in California due to the city's unique geography. Phoenix is a large urban area with no immediate surrounding cities, creating a "sky island" effect that intensifies the urban heat dome. This heat island effect, combined with the region's arid climate, creates unique challenges for understanding wildfire behaviour, air quality, and photochemistry in Arizona. These features make Arizona a distinctive study area, with broader implications for understanding wildfire impacts in other arid regions globally, such as parts of the Middle East, Australia, and northern Africa, where similar climatic conditions and urbanization patterns exist.

We have now included a discussion of this novelty in the introduction to highlight the significance of our work in this context.

Reference:

Greenslade, M., Guo, Y., Betito, G., Mirrezaei, M. A., Roychoudhury, C., Arellano, A. F., and Sorooshian, A.: On ozone's weekly cycle for different seasons in Arizona, Atmospheric Environment, 334, 120703, https://doi.org/10.1016/j.atmosenv.2024.120703, 2024.

2. *Section 3.1 The authors should simply explain the reasons for the selection of this period.*

Thank you for the comment. We have included an explanation in Section 2.1 detailing our reasons for selecting this period. Specifically, we chose June 2021 as it represents a period of intense fire activity in Arizona, driven by extreme drought and one of the hottest Junes on record, with temperatures exceeding 115°F (46°C) and no significant precipitation. These conditions contributed to episodes of $O_3$ exceedance under the influences of multiple large wildfires, including the Telegraph Fire and the Rafael Fire.

3. *Line 359: FNR often shows large uncertainties. Why threshold could you use to distinguish the VOC- or NOx-limited regions.*

Thank you for the comment. The concentration of formaldehyde (HCHO) serves as an indicator for volatile organic compound (VOC) reactivity as it exhibits a positive correlation with proxy radicals (Sillman, 1995). Sillman (1995) identified that elevated HCHO/NOy ratios typically indicate NOx-limited regimes, whereas reduced HCHO/NOy ratios are indicative of VOC-limited regimes. Martin et al. (2004) found that during summer, the transition between radical- and NOx-limited regimes occurs at a particular ratio threshold. Using the Community Multiscale Air Quality (CMAQ) model with finer resolution for the entire continental U.S., Duncan et al. (2010) proposed that formaldehyde-to-nitrogen oxides ratios (FNRs) below 1 suggest a VOC sensitivity regime, FNRs between 1 and 2 indicate a transition zone between VOC and NOx sensitivities, and FNRs above 2 are characteristic of a NOx-sensitive regime. It is important to note that variations in meteorological variables, emission sources, and pollution levels can alter the ozone production regime. In different studies, various FNR thresholds are calculated. i.e., satellite column retrievals of FNR of 0.7–2.3 in Schroeder et al. (2017), and 3.2–4.1 in Jin et al. (2020). In addition, Acdan et al. (2022) used ground-based PAMS measurements and suggested a FNR of 0.3–1.0 for transition over the Lake Michigan region. In our study, we are following Duncan et al. (2010) which linked FNR with surface $O_3$ sensitivity in model simulation and used in several studies (Tang et al., 2012; Jin and Holloway, 2015, Souri et al., 2017) by defining FNRs less than 1 as VOC sensitivity regime, FNRs between 1 and 2 as a transition between VOC and NOx sensitivities ('the transitional zone'), and FNRs greater than 2 as NOx-sensitive regime. Please see Guo et al. (2024) for more details.

Here we have included the discussion of FNR threshold in section 3.2 as follows:

"For the FNR threshold, we will adopt the same approach as Guo et al. (2024), following the methodology of Duncan et al. (2010), who linked the FNR with surface $O_3$ in model simulations. According to this framework, the sensitivity regime is defined as follows: when FNR is less than 1, it is classified as VOC-limited; values between 1 and 2 indicate a transitional regime; and an FNR greater than 2 indicates a $NO_x$-limited regime."

4. *The authors used H2O2/HNO3 to identify the oxidizing capacity of the atmosphere and the relative contributions of different chemical pathways to O₃ Please examine the predictive accuracy of H2O2 and HNO3 firstly. Besides, the threshold also shows large uncertainties. Please explain the detailed reasons.*

Thank you for your comment. In this study, the $H_2O_2/HNO_3$ ratio was employed as a complementary metric to reinforce and provide additional context for the chemical regime shifts primarily identified using the $HCHO/NO_2$ ratio, which serves as our primary indicator. The $H_2O_2/HNO_3$ ratio offers valuable insights into the oxidizing capacity of the atmosphere and the relative contributions of different chemical pathways to $O_3$ production. However, we acknowledge the uncertainties in its threshold values and emphasize that its use in this study is supplementary to the $HCHO/NO_2$-based chemical regime classification.

Regarding the threshold values for $H_2O_2/HNO_3$, there is indeed variability across studies. Sillman (1995) suggested a threshold of 0.4, where values above indicate a $NO_x$-limited regime, and values below indicate a VOC-limited regime. Subsequently, Sillman et al. (1997) proposed a lower threshold of 0.2, while Lu and Chang (1998) introduced a broader transition range of 0.8-1.2. Zhang et al. (2009) later recommended a higher threshold of 2.4 for summertime conditions. This variability underscores the uncertainties and region-specific applicability of these thresholds.

In this study, regardless of the specific thresholds used to define VOC-limited or $NO_x$-limited regimes, we focus on the changes in the $H_2O_2/HNO_3$ ratio in response to wildfire smoke. We have added a discussion in the manuscript highlighting the uncertainties associated with the $H_2O_2/HNO_3$ thresholds and clarify the supporting role of this ratio in interpreting the chemical regimes.

References

- Sillman, S., The use of NOy, $H_2O_2$, and $HNO_3$ as indicators for ozone-$NO_x$-hydrocarbon sensitivity in urban locations, J. Geophys. Res., 100, 14175–14188, 10.1029/94JD02953, 1995.

- Sillman, S., D. He, C. Cardelino, and R. E. Imhoff, The use of photochemical indicators to evaluate ozone-$NO_x$-hydrocarbon sensitivity: Case studies from Atlanta, New York, and Los Angeles, J. Air Waste Manage. Assoc., 47, 642–652, DOI: 10.1080/10962247.1997.11877500, 1997.

- Lu, C.-H., and J. S. Chang, On the indicator-based approach to assess ozone sensitivities and emissions features, J. Geophys. Res., 103, 3453–3462, doi:10.1029/97JD03128, 1998.

- Zhang, Y., Wen, X. Y., Wang, K., Vijayaraghavan, K., & Jacobson, M. Z., Probing into regional $O_3$ and particulate matter pollution in the United States: 2. An examination of formation mechanisms through a process analysis technique and sensitivity study, Journal of Geophysical Research: Atmospheres, 114(D22), https://doi.org/10.1029/2009JD011900, 2009.

5. *The limitations of this study should be added in the conclusion.*

Thank you for the suggestion. We have included the limitations in the Conclusion section 4 as:

"Our study has several limitations. First, we only tagged $NO_x$ emissions, which may not fully capture $O_3$ dynamics in VOC-limited regions where VOCs play a more significant role in $O_3$ formation. Additionally, the limited number of cases analyzed reduces the generalizability of the results. Expanding the study to include more cases would improve robustness. Furthermore, the absence of vertical profile data limits the depth of the analysis, particularly for understanding pollutant distributions in the upper atmosphere. In the future, with more spatial coverage, including diurnal variation data from Tropospheric Emissions: Monitoring of Pollution (TEMPO), would enhance the temporal and spatial representation of $O_3$ and aerosol concentrations. Refinements in fire emission estimates from FINN, plume rise calculations, and the validation of aerosol direct radiative effects on dynamics and thermodynamics, particularly radiation fluxes would increase model accuracy. Lastly, incorporating more detailed sectoral tags would better differentiate emission sources and improve $O_3$ and aerosol impact assessments. Moreover, since tagged $O_3$ is represented as tracers in the model, its production and immediate loss are primarily captured, while some loss processes may not be fully accounted for. Processes such as detailed fire-plume shading effects and nighttime $O_3$ titration chemistry may therefore be underrepresented."

---

## Referee Report (RR1)

Review of "Source contribution to ozone pollution during June 2021 in Arizona: Insights from WRF-Chem tagged O3 and CO"
By Guo et al

This paper presents a study on the source attribution of surface and tropospheric O3 and CO in Arizona during one month (June 2021) when fire emissions impacted the area. The authors use WRF-Chem tagged simulations as well as WRF-Chem sensitivity simulations (with and without fire) to determine the budget of O3 for the region, including the fire contribution. They also investigate the meteorological, diurnal, and chemical conditions that influence O3 production for two case studies in detail. While the study covers only a small spatial and temporal region/period, this kind of detailed case study is very interesting as there remains a high level of uncertainty and variability in O3 production during fire events.

The authors have revised the manuscript based on the first round of reviews, and the concerns raised in the reviews have nearly all been addressed. I have only one major concern remaining, related to the authors' interpretation of the tagged Mexico anthropogenic contribution when discussing Figure 9.

In Figure 9, I've assumed that the date tick marks correspond with 00:01 local time, thus, when the authors discuss June 15 and 17 in the text, they are referring to the particular area after (to the right) of the 0615 and 0617 ticks.
As such, it appears incorrect to say that on June 15th the Mexican Anthro contribution (red) exceeded the Arizona Anthro contribution (green), since this is only true in the first half (midnight to ~8am) of June 15[th]. The big peak when the exceedance (>70 ppb) occurred, has a large red area, indicating that the local anthropogenic contribution from Arizona is greater than that of Mexico, for a significant time period over this day, particularly during the exceedance time.

While the authors have revised the text in this discussion based on the first review, the new text is quite contradictory, e.g;
"exceedances of the NAAQS 70 ppb $O_3$ standard in Yuma are **often significantly influenced by emissions from Mexico**" yet "**these contributions are modest in absolute terms**".
And:
"While **not dominant** overall..." yet "these transboundary emissions **play a substantial role in elevating** $O_3$ above background levels and contribute to exceedances"

...It's as though by including the Mexican (green) portion at the top of the other contributions in the time series (Figure 9), they have perceived that the Mexican contribution caused the exceedances in Arizona. Whereas, if the Arizona (red) portion were placed at the top, one would perceive that *that* contribution caused the exceedance.

Therefore, I highly recommend that the text discussing Figure 9 be changed further to something like this before publishing:

"Figure 9 shows that $O_3$ levels in Yuma are largely dominated by the background level, primarily from long-range transport and natural sources. The exceedances of the NAAQS 70 ppb $O_3$ standard in Yuma were significantly influenced by a peak in this background contribution on June 15 and 17th when the background made up X% and Y%, respectively of the total daytime O3. On June 15 and 17th, the anthropogenic contributions from Arizona were W% and Z%, respectively, and the anthropogenic contributions from Mexico were U% and V% respectively." (filling in the percents from your tagged simulation data)

This new suggested text would rephrase the discussion of these results in a quantitative way and remove any perception bias from the interpretation of Fig 9. The authors can also add a brief sentence related to the fact that these are modelled results, and the modelled peaks on June 15th and 17th are +/-X% different from the measurement peaks.

I also have one minor comment:
Figures 11(c and d) and 13 (c and d): When showing differences ("Diff"), please change to a divergent colour scale (e.g. blue to red with white at zero) instead rainbow, which makes it difficult to understand the results.

---

## Author Response (AR2)

Response to Reviewers Comments.

**Reviewer 1: General Comment**

*Authors have revised the manuscript based on the first round of reviews, and the concerns raised in the reviews have nearly all been addressed.*

**Response:**

We appreciate the reviewer's general positive comments on our revisions.

**Reviewer 1: Main Comment**

*I have only one major concern remaining, related to the authors' interpretation of the tagged Mexico anthropogenic contribution when discussing Figure 9. In Figure 9, I've assumed that the date tick marks correspond with 00:01 local time, thus, when the authors discuss June 15 and 17 in the text, they are referring to the particular area after (to the right) of the 0615 and 0617 ticks. As such, it appears incorrect to say that on June 15th the Mexican Anthro contribution (red) exceeded the Arizona Anthro contribution (green), since this is only true in the first half (midnight to ~8am) of June 15th. The big peak when the exceedance (>70 ppb) occurred, has a large red area, indicating that the local anthropogenic contribution from Arizona is greater than that of Mexico, for a significant time period over this day, particularly during the exceedance time. While the authors have revised the text in this discussion based on the first review, the new text is quite contradictory, e.g; "exceedances of the NAAQS 70 ppb O₃ standard in Yuma are often significantly influenced by emissions from Mexico" yet "these contributions are modest in absolute terms". And:*

*"While not dominant overall…" yet "these transboundary emissions play a substantial role in elevating O₃ above background levels and contribute to exceedances"…It's as though by including the Mexican (green) portion at the top of the other contributions in the time series (Figure 9), they have perceived that the Mexican contribution caused the exceedances in Arizona. Whereas, if the Arizona (red) portion were placed at the top, one would perceive that that contribution caused the exceedance. Therefore, I highly recommend that the text discussing Figure 9 be changed further to something like this before publishing:*

*"Figure 9 shows that O₃ levels in Yuma are largely dominated by the background level, primarily from long-range transport and natural sources. The exceedances of the NAAQS 70 ppb O₃ standard in Yuma were significantly influenced by a peak in this background contribution on June 15 and 17th when the background made up X% and Y%, respectively of the total daytime O3. On June 15 and 17th, the anthropogenic contributions from Arizona were W% and Z%, respectively, and the anthropogenic contributions from Mexico were U% and V% respectively." (filling in the percents from your tagged simulation data). This new suggested text would rephrase the discussion of these results in a quantitative way and remove any perception bias from the interpretation of Fig 9. The authors can also add a brief sentence related to the fact that these are modelled results, and the modelled peaks on June 15th and 17th are +/-X% different from the measurement peaks.*

**Response:**

We appreciate the reviewer's keen observation and thoughtful suggestions for rewording the interpretation of Figure 9. We have carefully revised the paragraph accordingly. It now reads:

"The exceedances of the NAAQS 70 ppb $O_3$ standard in Yuma were significantly influenced by a peak in this background contribution on June 15th and 17th when the background made up ~65% and ~70%, respectively, of the total daytime $O_3$. On June 15th and 17th, the anthropogenic contributions from Arizona were 20% and 10%, respectively, and the anthropogenic contributions from Mexico were 8% and 13% respectively. We note, however, that these are modeled results, and the modeled peaks on June 15th and 17th are 16 to 30% different from the measurement peaks, overestimating on June 15th and underestimating on June 17th."

**Reviewer 1: Minor Comment**

*Figures 11(c and d) and 13 (c and d): When showing differences ("Diff"), please change to a divergent colour scale (e.g. blue to red with white at zero) instead rainbow, which makes it difficult to understand the results.*

**Response:**

We thank the reviewer for this comment. We have revised Figure 11c-d and 13 c-d to have a divergent color scale.

**Reviewer 2: General Comments**

*Although the author make significant revisions about this manuscript. The conclusion is still well-known based on previous studies. The authors cannot treat a typical region as the novelty of this study. Arizona did not show unique characteristic as Tibetan Plateau or Antarctic. Furthermore, it is also necessary to obtain new findings even if the study domain is located on special regions. The authors should summarize more novel findings compared with previous studies. Moreover, the authors did not examine the accuracy of H2O2 and HNO3 simulations. Therefore, I cannot recommend the manuscript for publication on ACP.*

**Response:**

We sincerely appreciate the reviewer's thoughtful feedback on our revisions, as well as the recognition of the lack of novelty of this study and the accuracy of $H_2O_2$ and $HNO_3$ simulations. In response, we have made substantial revisions across multiple sections (incl. title) to better highlight the scientific contributions and unique aspects of our work, ensuring that these key elements are clearly conveyed to the readers.

We have revamped our Abstract, Introduction, Main Sections, and Conclusions. We added a good number of references to provide context to some of our findings. Specifically, we have made the following key improvements:

1. **Enhanced focus on source attribution using advanced modeling:**

We have clarified our goal to quantify fire contributions to $O_3$ levels using a high-resolution, coupled weather-air quality model with tagging capabilities. To our knowledge, this is the first study to apply such a framework at this resolution, as most previous tagged $O_3$ studies relied on global models (CTMs, not ESMs) or coarser-resolution regional/offline models that lack chemistry-meteorology feedbacks. This is particularly important for our case study on a compound event in a complex terrain environment, where coarser models may underestimate source attribution accuracy and fail to capture feedback mechanisms. We have emphasized this in the revised manuscript.

2. **Refined background $O_3$ estimates for Arizona and the Southwest U.S.:**

Our study presents an alternative estimate of background $O_3$ in Arizona, which can be extrapolated to the broader Southwest U.S. While our results are consistent with recent observational studies (notably Parrish et al., 2025), our model's finer resolution allows us to capture spatial variations in background $O_3$—an essential factor in accurate source attribution. We have expanded the discussion on this topic and included an additional figure to illustrate these variations.

3. **Fire-induced wind modifications and their role in source attribution:**

Our analysis reveals that wildfires altered local wind patterns, an essential finding that must be considered in source attribution for compound events. We have expanded this discussion with additional figures illustrating the impact of fires on wind.

**4.    Comprehensive analysis of smoke plume chemistry:**

We provide a detailed illustration of the spatial distribution and cross-section of $O_3$ precursors and chemical regimes within the smoke plume. This is a direct advantage of our coupled modeling framework, which we argue is more realistic than photochemical box models, as it accounts for environmental variations and chemistry-meteorology feedbacks—a capability missing in models like HYSPLIT. We have further strengthened this section by comparing our results with available observations and discussing agreements with plume-based (box model) studies.

**5.    Model comparison with ground and satellite observations:**

We have compared our model results with as many available ground-based and satellite-based measurements as possible. In combination with the improvements above, we argue that this study is one of the few that integrates multiple strengths to enhance the accuracy and robustness of wildfire-related $O_3$ assessments.

**6.    Expanded chemical regime analysis and refinement of comparison metrics:**

We have expanded the chemical regime analysis by incorporating a comparison with TROPOMI FNR. Additionally, we have removed the discussion on $H_2O_2/HNO_2$, given the lack of available observations for validation. Instead, we replaced it with $O_3/NO_x$ comparisons and introduced a new figure comparing TROPOMI-derived columnar FNR with WRF-Chem simulations (with and without fire emissions).

**7.    Broader applicability to arid regions with persistent $O_3$ pollution:**

While our focus is on Phoenix, Arizona, we strongly believe that the insights gained from this study are applicable to other arid environments, particularly those struggling with persistent $O_3$ pollution. The $O_3$ dynamics in such regions—especially those with large urban centers—remain poorly studied and insufficiently observed. We have highlighted these broader implications in our revised manuscript.

These revisions strengthen the clarity, impact, and scientific contributions of our study, reinforcing its novelty and relevance in wildfire-driven $O_3$ research and highlighting the utility of this approach in future studies.